

# Comparing contact and immersion freezing from continuous flow diffusion chambers

Baban Nagare[1], Claudia Marcolli[1,2], André Welti[1,3], Olaf Stetzer[1], and Ulrike Lohmann[1]

[1]Institute for Atmospheric and Climate Science, ETH, Zurich Switzerland
[2]Marcolli Chemistry and Physics Consulting GmbH, Zurich, Switzerland
[3]Leibniz Institute for Tropospheric Research, Leipzig, Germany

*Correspondence to:* Claudia Marcolli (claudia.marcolli@env.ethz.ch)

**Abstract.** Ice nucleating particles (INPs) in the atmosphere are responsible for glaciating cloud droplets between 237 K and 273 K. Different mechanisms of heterogeneous ice nucleation can compete under mixed-phase cloud conditions. Contact freezing is considered relevant because higher ice nucleation temperatures than for immersion freezing for the same INPs were observed. It has limitations because its efficiency depends on the number of collisions between cloud droplets and INPs. This study compares immersion and contact freezing efficiencies of three different INPs. The contact freezing data was obtained with the ETH CoLlision Ice Nucleation CHamber (CLINCH) using 80 μm diameter droplets which can interact with INPs for residence times of 2 s and 4 s in the chamber. The contact freezing efficiency was calculated by estimating the number of collisions between droplets and particles. Theoretical formulations of collision efficiencies gave too high freezing efficiencies for all investigated INPs, namely AgI particles with 200 nm electrical mobility diameter, 400 and 800 nm diameter ATD and kaolinite particles. Comparison of freezing efficiencies by contact and immersion freezing is therefore limited by the accuracy of collision efficiencies. The concentration of particles was 1000 cm$^{-3}$ for ATD and kaolinite and 500, 1000, 2000 and 5000 cm$^{-3}$ for AgI. For concentrations < 5000 cm$^{-3}$, the droplets collect only one particle on average during their time in the chamber. For ATD and kaolinite particles, contact freezing efficiencies at 2 s residence time were smaller than at 4 s, which is in disagreement with a collisional contact freezing process but in accordance with contact freezing inside-out or immersion freezing. For best comparison with contact freezing results, immersion freezing experiments of the same INPs were performed with the continuous flow diffusion chamber IMCA/ZINC for 3 s residence time. In IMCA/ZINC, each INP is activated into a droplet in IMCA and provides its surface for ice nucleation in the ZINC chamber. The comparison of contact and immersion freezing results did not confirm a general enhancement of freezing efficiency for contact compared with immersion freezing experiments. For AgI particles the onset of heterogeneous freezing in CLINCH was even shifted to lower temperatures compared with IMCA/ZINC. For ATD, freezing efficiencies for contact and immersion freezing experiments were similar. For kaolinite particles, contact freezing became detectable at higher temperatures than immersion freezing. Using contact angle information between water and the INP, it is discussed how the position of the INP in or on the droplets may influence its ice nucleation activity.



## 1 Introduction

Aerosol particles influence the climate system in different ways. They scatter or absorb the incoming solar radiation or can absorb thermal radiation. Aerosol particles also play a role in cloud formation by acting as cloud condensation nuclei (CCN) and as ice nucleating particles (INPs). INPs help in ice nucleation depending on their physical and chemical properties, temperature of the environment and presence of supercooled droplets. Heterogeneous ice formation may take place with the help of INPs between 237 K and 273 K in the mixed-phase clouds regime. In this regime, three pathways of ice nucleation are differentiated, namely, immersion freezing, condensation freezing and contact freezing. In condensation freezing, water vapor condenses on the INP at temperatures < 273 K to form a liquid droplet which freezes instantaneously. Immersion freezing (IF) takes place when an INP acts as CCN and the formed droplet freezes when the temperature is lowered. In contact freezing the INP collides with the droplet followed by freezing. Contact freezing (CF) in the original sense is defined as the process in which freezing of a supercooled droplet results from the collision with an aerosol particle (Ladino Moreno et al., 2013; Vali, 1985). This view of contact freezing has been challenged by Durant and Shaw (2005) who discriminate between traditional contact freezing as collisional contact freezing where ice nucleation is due to the collision itself and contact freezing inside-out, in which an INP in contact with the water-air interface of a droplet, from either the inside or the outside of the droplet nucleates ice at a higher temperature than when it is totally immersed in the droplet (Durant and Shaw, 2005; Gurganus et al., 2014; Fornea et al., 2009; Murray et al., 2012; Shaw et al., 2005).

Contact freezing is probably the least understood pathway of ice formation in the atmosphere. The collision rate between particles and droplets depends on the collision efficiency which is not well determined in the accumulation mode size range as pointed out by Nagare et al. (2015). Collision efficiency ($CE$) is defined as the fraction of particles in the cylindrical volume swept out by a falling droplet that collide with it. $CE$ depends mainly on particle size and droplet size. Other factors which influence $CE$ are relative humidity of the environment and charges on droplets and particles. For Aitken mode particles, Brownian motion of the particles is usually the dominant collision process and $CE$ can exceed 1 because of the high mobility of the particles in this size range. For coarse mode particles, impaction and interception of particles are the dominant contributors to $CE$. The minimun of $CE$ (Greenfield gap) is in the accumulation mode where thermophoresis and diffusiophoresis may also contribute to $CE$ (Nagare et al., 2015).

Ladino Moreno et al. (2013) have reviewed different theoretical models and experimental studies on contact freezing and also pointed out large discrepancies in the available experimental data. Quantification of the number of INPs required for contact freezing, time dependence of contact freezing, dependence on particle type and size were listed amongst the most uncertain parameters. Hoffmann et al. (2013a, b), and Niehaus et al. (2014) attempted to quantify the number of INPs required to freeze a droplet by contact freezing for their respective experiments. In Hoffmann et al. (2013a, b), an highly electrified droplet is suspended in an electrodynamic balance. The freezing probability of a droplet on a single collision was shown to be a steep function of temperature. Nine collisions were necessary to freeze the droplet at 244 K while a single collision was sufficient to freeze the droplet at 239 K for illite particles with a mobility diameter of 750 nm. They conclude that contact freezing is the dominant mechanism over immersion freezing (Hoffmann et al., 2013a). Niehaus et al. (2014) found that for mineral





dust from different origins $10^3$ to $10^5$ particles had to collide with a droplet deposited on the glass slide in the temperature range of 253 K to 258 K. Contact freezing and immersion freezing have been previously compared by Levin and Yankofsky (1983) for bacterial cells where onset temperatures for contact freezing were shown to be 2 K higher. However, they did not state how many particles were needed to collide with the droplets to initiate freezing. Also Ladino et al. (2011) concluded that there are some hints that contact freezing is more efficient than immersion frezing for kaolinite particles. They attempted to derive the freezing efficiency per single particle using theoretical formulations of collision efficiencies to calculate the number of collisions between droplets and particles and obtained unrealistic freezing efficiencies on the order of 10 to 100 for 26 μm diameter droplets and 400 nm kaolinite particles. The too large values of the freezing efficiency were attributed to the overestimation of droplet size in calculating $CE$. They also were mistaking liquid droplets as frozen droplets because multiple droplets were simultaneously present in the laser beam of the detector (Ladino Moreno et al., 2013).

In order to compare the efficiency of immersion freezing and contact freezing, we performed a series of experiments with silver iodide, kaolinite and Arizona Test Dust (ATD) in immersion and contact freezing mode. Silver iodide is known to be a very good ice nucleus (Vonnegut, 1949) inducing ice nucleation up to 269 K, while ATD and kaolinite become efficient ice nuclei only at lower temperatures. Silver iodide has been reported to be more efficient as IN in contact than in immersion mode (DeMott, 1995). Kaolinite and ATD have been widely tested in laboratory studies as immersion freezing nuclei. Kaolinite is a clay mineral and accounts for 13 % of dust mass in the atmosphere (Atkinson et al., 2013). It has been studied previously in immersion freezing (e.g., Welti et al., 2012) and contact freezing studies (e.g., Ladino et al., 2011; Svensson et al., 2009). ATD has been previously studied for immersion freezing by Marcolli et al. (2007) and Niedermeier et al. (2010) and for contact freezing by Niehaus et al. (2014). ATD is composed of quartz, feldspar, carbonate, illite, kaolinite and other clays (Broadley et al., 2012).

## 2   Experimental setups and procedures

### 2.1   Instrument description

#### 2.1.1   CLINCH setup

Contact freezing data was obtained with the ETH CoLlision Ice Nucleation CHamber (CLINCH). This instrument has been used previously by Ladino et al. (2011) for a contact freezing study with kaolinite as INPs. In CLINCH, aerosol and water droplets collide and may freeze by contact. The extension of the chamber length from 40 cm used by Ladino et al. (2011) to 80 cm for the current study makes it possible to observe the frozen fraction of droplets ($FF$) i.e. the ratio of number of frozen droplets to total number of droplets, at residence times of 2 s and 4 s with 80 μm diameter droplets compared to 26 μm droplets used by Ladino et al. (2011). The droplet diameter is changed from 26 to 80 μm in order to increase the geometrically swept out volume by the droplet and to avoid significant change in the droplet size due to evaporation in the chamber. $80 \pm 3$ μm diameter droplets are generated with a droplet generator (Ulmke et al., 2001) at the top center of the chamber with a frequency of 100 Hz. The droplets are generated with pure water (Milli-Q, 18.2 MΩ) at a temperature of 281 K. The relaxation time for





a droplet to reach its terminal velocity ($0.186 \ \mathrm{ms}^{-1}$) is 0.2 s and the time needed to reach the target temperature is about 0.1 and 0.6 s when the chamber is kept at 261 and 235 K, respectively (Nagare et al., 2015). While performing the experiment, the walls of the chamber are coated with a thin layer of ice creating an ice saturated environment inside the chamber.

Aerosol particles enter the chamber at the top in air streams from both sides with a flow velocity of 1 LPM and can interact with the droplets inside the chamber. $FF$ can be determined with the in-house developed Ice Optical DEtector (IODE) (Nicolet et al., 2010; Lüönd et al., 2010) which discriminates water droplets from ice crystals by measuring the depolarization of the backscattered light of a laser beam. In order to avoid the presence of several droplets simultaneously in the laser beam, a new laser was installed (402 nm, Schaefter + Kirchhoff laser Makroliniengenerator13LTM) providing a rectangular instead of a circular laser beam. At each temperature, a blank experiment without aerosol particles was performed before the aerosol stream was turned on. A more detailed description of the instrument and experiment is given in Nagare et al. (2015).

### 2.1.2 IMCA/ZINC setup

Immersion freezing experiments were performed using the IMCA/ZINC setup (Welti et al., 2012). This setup combines the Zurich Ice Nucleation Chamber (ZINC) (Stetzer et al., 2008) with the vertical extension Immersion Mode Cooling chAmber (IMCA) (Lüönd et al., 2010). In brief, the aerosol particles are activated as CCN in the IMCA part at a relative humidity with respect to water > 120 % and temperature > 300 K. These activated droplets are then cooled down in the IMCA part and reach the target temperature for freezing when they enter the ZINC chamber. The droplets are $18 - 20 \ \mu\mathrm{m}$ in diameter when they leave the IMCA part and enter the water saturated environment in ZINC which is created by ice coatings on the parallel walls, which are kept at different temperatures. $FF$ can be determined using the depolarization detector IODE at different residence times from 1 s to 21 s. A more detailed description of the instrument and experiment is given in Welti et al. (2012). Characteristics of the IMCA/ZINC and the CLINCH experiments are compared in Table 1.

### 2.1.3 Aerosol generation and sampling

Silver iodide was precipitated by mixing 0.1 M solutions of potassium iodide and silver nitrate. The aerosol particles were generated by atomizing this suspension and dried (for details refer to Nagare et al. (2015)). The suspension was usually prepared the day before a measurement series was started and used for a measurement series performed during typically 2 days. Between measurements the suspension was kept in the dark. Kaolinite (Fluka, Sigma Aldrich GmbH) and ATD (Powder Technology Inc.) particles were aerosolised in a fluidized bed aerosol generator (TSI Model 43400A). The aerosol stream was passed through a cyclone to remove large particles. Aerosol particles were selected based on their electrical mobility with a Differential Mobility Analyzer (DMA TSI 3081) with an upstream impactor. These size selected particles were used for either contact or immersion freezing experiments in the respective experimental setups. The concentration of particles in CLINCH was measured at the end of the chamber using a condensation particle counter (CPC, TSI 3772).



## 3 Experimental results

Figure 1 shows the $FF$ observed for silver iodide as INP in CLINCH (triangles) as a function of chamber temperatures for droplet residence times of 2 s in panel (a) and 4 s in panel (b) for different concentrations of silver iodide. The gray shaded area is the experimentally determined homogeneous freezing regime of droplets in CLINCH from blank experiments. The black horizontal line marks the lower reliability limit of differentiation between ice and water determined from blank experiments. As the temperature of the chamber decreases to < 250 K, the $FF$ starts to rise and then remains constant. The frozen fraction due to immersion freezing from IMCA/ZINC experiments with 3 s residence time in the ZINC chamber is shown as circles for comparison with contact freezing. Silver iodide particles produced by our method are found to be much more efficient INP in terms of onset temperature in immersion than in contact freezing mode. The onset temperature for silver iodide particles as INP is 265 K while for contact freezing significant frozen fractions were observed only below 250 K except for the highest concentration and 4 s residence time. For the highest concentration used in our experiment, the onset temperature for contact freezing is 258 K.

Figure 2 shows the frozen fraction of droplets when ATD was used as INP in contact and immersion freezing mode. The frozen fraction due to immersion freezing shown are for 800 nm particles and 3 s residence time in the ZINC chamber. There is no significant difference in onset temperature for immersion and contact freezing for ATD. Figure 3 shows the frozen fraction for experiments performed with kaolinite. For this INP, the onset temperature of contact freezing is 3 K higher than for immersion freezing. Possible reasons for this will be discussed in Sect. 5.5 and 5.6.

## 4 Freezing efficiencies

### 4.1 Calculation of freezing efficiency from frozen fraction

The frozen fraction measured by CLINCH depends on the collision efficiency and the freezing efficiency. For a further evaluation and comparison of contact freezing and immersion freezing $FF$ has to be converted to $FE$ as follows:

$$FE = \frac{FF}{N} \tag{1}$$

where $N$ is the number of collisions for a droplet with the aerosol particles and can be calculated as (Ladino et al., 2011)

$$N = CE \times C \times L \times \pi \times (R + r)^2 \tag{2}$$

where $C$ is the concentration of the particles, $R$ and $r$ are the radii of droplets and particles, respectively, and $L$ is the effective length experienced by the droplet given as

$$L = \frac{U(R)l}{U(R) + V_{flow}} \tag{3}$$

where $l$ is the geometrical length traced by the droplet, $U(R)$ is the terminal velocity of the droplet and $V_{flow}$ is the flow velocity of the carrier gas in CLINCH.





Eq. (1) assumes that the droplet has collected $N$ particles and freezes due to the last particle that it has collected. However, the droplet can freeze on collision with the first particle and then collect other particles. Assuming that one collision is enough for ice nucleation, leads to the following expression for $FE$:

$$FE = \frac{FF}{1 - e^{-N}}. \tag{4}$$

Here the denominator indicates the fraction of unfrozen droplets after N collisions with the particles.

Since $FE$ is derived by normalizing $FF$ with respect to $N$, $FE$ should be independent of the residence time when freezing occurs at the first contact of a particle with a droplet. If freezing efficiencies of 2 s ($FE(2s)$) and 4 s residence times ($FE(4s)$) are the same within the experimental uncertainty, this can be considered as an indication of immediate freezing when the first particle collides with a droplet. Conversely, $FE(4s) > FE(2s)$ suggests that collisions with more than one particle are
needed to freeze a droplet or that freezing is not immediate when a particle hits a droplet but that more time is needed on average. Such a time dependent freezing process would be in accordance with an immersion freezing mechanism assuming that the droplet only freezes when the particle becomes immersed in it or with contact freezing inside-out when the particle adheres to the surface of the droplet. If $N < 1$, it is unlikely that $FE$ is influenced by the number of collisions and we will interpret $FE(2s) = FE(4s)$ as a criterion for collisional contact freezing and $FE(4s) > FE(2s)$ as a criterion for freezing
in immersion mode or contact freezing inside-out.

### 4.2 Freezing efficiency of silver iodide particles

We derived a collision efficiency $CE = 0.13$ for 200 nm diameter AgI particles with 80 μm droplets in our previous study (Nagare et al., 2015). This number is an order of magnitude higher than the values calculated with commonly used theoretical formulations of collision efficiencies. Figure 4 shows $FE$ of 200 nm diameter silver iodide particles for droplet residence times
of 2 s (open symbols) and 4 s (filled symbols) calculated using Eqs. (1) (panel a) and (4) (panel b). The number of collisions for the different particle concentrations range between 0.1 and 2.35 as listed in Table 3. Panel (a) of Fig. 4 shows that $FE$ does not exceed 0.5 for $C = 5000\,\mathrm{cm}^{-3}$ because it is assumed that on average 2.35 collisions are necessary to freeze a droplet. This led us to Eq. (4) to calculate $FE$, which assumes that already the first collision induces droplet freezing. Panel (b) of Fig. 4 shows that this assumption leads to a grouping of $FE$ data around 1 for T < 245 K for all particle concentrations
and residence times of 2 s and 4 s. This reinforces the assumption that the first contact leads to droplet freezing in this temperature range and confirms the plateau condition used in Nagare et al. (2015) to derive $CE$. Above this temperature, $FE$ values significantly different from zero are only reached for concentrations of 5000 $\mathrm{cm}^{-3}$. This would imply that above this temperature more than one collision is necessary for contact freezing. For T < 245 K, $FE$s for 2 s and 4 s residence times are the same within measurement uncertainties suggesting that ice nucleation occurs immediately when the particle hits the
droplet. At higher temperatures the data points are quite scattered impeding a clear conclusion. Also shown are immersion freezing measurements with IMCA/ZINC with AgI particles that were prepared the same way as the ones for the contact freezing experiments. Residence time in the ZINC chamber was 3 s. Surprisingly, the onset temperature and the efficiency for immersion freezing are significantly higher than for contact freezing. This observation is further discussed in Sect. 5.4.





### 4.3 Freezing efficiency of ATD particles

Figure 5 shows $FE$ in contact freezing mode for 800 nm ATD particles calculated using Eq. (1). $N$ is calculated based on 4 different assumptions for $CE$. For panel (a), $CE = 0.0033$ from the theoretical formulation by Park et al. (2005) and Wang et al. (1978) was used. Details of the calculations are given in Nagare et al. (2015), where discrepancies between theoretical calculations and experimental observations have been addressed. For 200 nm silver iodide particles, the experimentally deter-mined $CE$ is 14 times higher than the calculated one. In order to adjust $FE$ better to the theoretical upper limit of $FE = 1$ and due to lack of other available experimental values, the calculated collision efficiency was multiplied with the factor of 14 to calculate $N$ for panel (b). For a lower limit the experimentally derived $CE$ for 200 nm AgI particles ($CE = 0.13$) has been used to calculate $N$ in panel (c). In panel (d) $CE = 0.061$ was used which shifts $FE$ of 800 nm ATD particles close to 1, which is in accordance with the assumption that each collision leads to droplet freezing. The calculated $CE$ shown in panel (a) leads to unrealistically high $FE$. This was also observed by Ladino et al. (2011). Using a correction factor of 14 for $CE$, still yields $FE$ values $> 1$. Panels (c) and (d) give best estimates of lower and upper limits of freezing efficiency. The number of collisions for the different assumptions of $CE$ is listed in Table 2. For the lower limit case with $CE = 0.13$, $FE$ reaches values up to 0.5 for data points that can be unambiguously assigned to heterogeneous freezing. Contact freezing experiments do not show significantly different onset temperatures compared with immersion freezing experiments carried out with the IMCA/ZINC setup, where every droplet contains one particle. The active site parameterization developed by Marcolli et al. (2007) based on DSC experiments is shown as brown line in Fig. 5. It agrees well with the immersion freezing experiments carried out with 800 nm particles in the IMCA/ZINC chamber. Taking $CE = 0.061$ (panel d), contact freezing might be slightly more efficient than immersion freezing. Taking $CE = 0.13$ (panel c), contact freezing and immersion freezing seem to be similarly efficient. We did not convert the frozen fraction of 400 nm ATD particles to freezing efficiency because $FF$ is close to the detection limit. For all assumed values of collision efficiencies, the freezing efficiency at 4 s residence time is almost twice the value at 2 s res-idence time. As listed in Table 2 the number of collisions is $< 1$ for both residence times. While the uncertainty associated with the measurements at 2 s is quite large, this data still seems significantly lower than the 4 s residence time freezing efficiencies. An increasing freezing efficiency with increasing residence time is expected for immersion freezing (Hoffmann et al., 2013b; Welti et al., 2012) and contact freezing inside-out. Therefore, it is likely that freezing occurs due to one of these mechanisms rather than collisional contact freezing.

### 4.4 Freezing efficiency of kaolinite particles

Figure 6 shows freezing efficiency for 800 nm diameter kaolinte particles for 2 s (open triangles) and 4 s (filled triangles) residence times. The frozen fraction measured for 400 nm particles was not significant, therefore, we do not convert this data to freezing efficiency. Panels (a), (b) and (c) use three different assumptions to calculate $N$ as explained in the previous section for ATD. Shown as brown circles in Fig. 6 are the immersion freezing results of 800 nm Fluka kaolinite particles for 3 s residence time in the ZINC chamber. Freezing efficiencies are in good agreement with the previously published $\alpha$-pdf parameterization by Welti et al. (2012) derived from immersion freezing experiments performed with the same setup (brown





line). In Wex et al. (2014) immersion freezing experiments with 700 nm kaolinite (Fluka) particles were performed with LACIS (shown as blue diamonds) and with a CFDC (shown as green diamonds). In the LACIS instrument, INPs are activated to droplets at $T = 257 - 260$ K while cooling to the targeted temperature. They are at the experimental temperature during 1.6 s while they evaporate. This lower residence time may explain the lower freezing efficiency observed in LACIS compared with

IMCA/ZINC. Slightly higher freezing efficiencies than in LACIS but still lower than in IMCA/ZINC were observed for 700 nm Fluka kaolinite particles in the CFDC (orange stars: Tobo et al. (2012); green diamonds: (Wex et al., 2014)). The contact freezing efficiencies from CLINCH are clearly higher for the lower limit of $CE = 0.046$ and slightly higher for the upper limit of $CE = 0.13$.

## 5 Discussion

### 5.1 Collision efficiency

Collision efficiency is a crucial parameter for an accurate comparison of contact and immersion freezing. Figures 5 and 6 show that freezing efficiencies of ATD and kaolinite particles calculated with theoretical formulations of $CE$ are at least by one order of magnitude too high. This corroborates the finding by Nagare et al. (2015), that $CE$ formulations need to be reassessed for temperature below 273 K. More such studies for different particle and droplet sizes are needed to improve the data base

for validation of calculated collision efficiencies at subzero temperatures, subsaturation with respect to water and droplets and particles with known charges. For measurements with the AgI aerosol, $FE = 1$ could be assumed for data points at T < 245 K, because they showed constant frozen fractions and IMCA/ZINC experiments determined $FE = 1$ at T < 245 K. For ATD and kaolinite, there was no temperature range where freezing occurred with an efficiency of one. Therefore, only upper and lower limits of collision efficiency can be estimated. This limits the comparison of contact with immersion freezing. Collision

efficiency is also a crucial factor to quantify the lifetime of the accumulation mode aerosol in the atmosphere because their lifetime strongly depends on the scavenging rate of particles by the droplets (Seinfeld and Pandis, 2006).

### 5.2 Contact freezing process

Various theoretical mechanisms underlying contact freezing have been proposed as explanations for the higher freezing efficiency in contact compared with immersion mode. They have been reviewed by Ladino Moreno et al. (2013). Here we discuss

them in brief. Cooper (1974) proposed that ice embryos formed on INPs in vapor are able to nucleate supercooled water upon collision with a droplet. His explanation relies on the classical nucleation theory and is based on the fact that the critical radius of an ice embryo for deposition nucleation is about 4-5 times larger than that for immersion freezing. Therefore a particle inactive as a deposition nucleus in the vapor, may nevertheless possess ice embryos larger than the critical size for an embryo immersed in water on it's surface. Such an embryo may induce freezing when immersed in water. This mechanism was refuted

by Fukuta (1975b). Fukuta (1975a) proposed a similar mechanism but with subtle differences. Similar to Cooper (1974), Fukuta (1975a) assumed that subcritical ice clusters form on the particles by vapor deposition. However, he rejected that these clusters





remain active, once they are immersed in the droplet as proposed by Cooper (1974). Instead, he assumed that freezing occurs during the wetting process when the water front moves over the particle, because this process gives rise to a transient high free energy zone which facilitates nucleation. This process should be only valid for hydrophobic nuclei. While the older theories focus on a collisional contact freezing mechanism, the more recent ones concentrate on contact freezing inside-out. Indeed,

experimental studies by Shaw et al. (2005) and Fornea et al. (2009) have shown that an INP that is not completely immersed in the droplet can trigger ice nucleation at higher temperatures. From simulations, Sear (2007) found that the nucleation rate is 4 orders of magnitude higher along the contact line where the water surface meets the surface of the particle. Based on classical nucleation theory, he considered this result as generic. Suzuki et al. (2007) found from their experiments with water droplets on silicon surfaces coated with various silanes that the temperature at which nucleation occurs at a contact line depends on the

contact angle between water and the substrate. On the other hand, Gurganus et al. (2011, 2013) investigated the freezing of droplets deposited on clean and coated silicon wafers and did not observe any preference of nucleation at the contact line. The same group also studied this phenomenon on catalyst substrates with imposed surface structures and found that the preferred nucleation site was the contact line in the case of nanoscale texture but not for microscale texture (Gurganus et al., 2014). Djikaev and Ruckenstein (2008) proposed that the line tension associated with the three phase contact line may indeed play

an important role. Whether contact freezing will be enhanced should depend on the interplay between the surface tensions and line tensions involved in this process.

The higher $FEs$ for 4 s than for 2 s residence time of the CLINCH experiments with ATD and kaolinite are in agreement with contact freezing inside-out or immersion freezing. This indicates that collision itself does not increase $FE$ but there seems to be an effect whether the INP adheres to the water surface or is immersed in the droplet. The situation is less clear for AgI.

For contact freezing experiments $FE$ at 2 s is the same as for 4 s residence time within error when on average one AgI particle or less collides with the droplet in the chamber. This result is in agreement with a collisional contact freezing mechanism, but may also result from a very high nucleation rate of immersion freezing and/or contact freezing inside-out at the investigated temperature.

### 5.3 Wetting of particles

When an insoluble or slightly soluble particle acts as cloud condensation nucleus (CCN), it is usually assumed that it becomes totally immersed into the droplet. However, whether the particle adheres to the droplet surface or becomes totally immersed depends on the wetting behavior of the particle. The wetting behavior can be quantified by the contact angle $\alpha$, which is related to the surface tensions of water with air $\sigma_{LA}$ and solid with air $\sigma_{SA}$ and the interfacial tension between solid and water $\sigma_{SL}$ through the Young equation (Hołownia et al., 2008) as follows:

$$\cos\alpha = \frac{\sigma_{SA} - \sigma_{SL}}{\sigma_{LA}}, \tag{5}$$





The change in surface tension when the particle that adheres to a surface of the droplet becomes totally immersed in the droplet is given as (Hołownia et al., 2008)

$$\Delta\sigma = \sigma_{SL} - \sigma_{SA} + \sigma_{LA} \tag{6}$$

Using Young's equation the change of surface tension is

$$\Delta\sigma = \sigma_{LA}(1 - \cos\alpha). \tag{7}$$

The change in the interfacial energy is given by

$$\Delta G = \Delta\sigma A \tag{8}$$

where $A$ is the surface area of the particle exposed to air when the particle adheres to the surface of the droplet. Considering a cubic particle and neglecting the curvature of the droplet, the area to be immersed in the droplet would be the area of one face of the cube. The particle will immerse in the droplet for negative $\Delta G$ and will remain on the surface for positive $\Delta G$. As can be seen from Eq. 8, $\Delta G$ is always positive and becomes zero for $\alpha = 0°$. This means that in the absence of other forces, complete wetting of the particle surface by water is needed for total immersion of the particle into the droplet.

### 5.4 Silver iodide

For AgI particles, freezing efficiencies for 2 s and 4 s residence times are the same within experimental uncertainties, which is in accordance with immediate freezing after collision. However, contact freezing inside-out and immersion freezing cannot be excluded, if these processes occur at a high rate. Whether AgI adheres to the surface after collision or becomes totally immersed depends on the contact angle between water and the AgI surface. Billett et al. (1976) observed a dependence of the contact angle on the silver concentration in the solution. For silver iodide prepared in stoichiometric ratio, they determined $\alpha = 45° - 50°$ for the intermediate advancing angle. We observed that most of the AgI particles adhered to the surface when we sprinkled them gently on water. This is in accordance with observations by Gokhale and Goold (1968) and Gokhale and Lewinter (1971). We therefore assume that silver iodide particles remain on the droplet surface after collision in the CLINCH chamber. It is also possible that the AgI particles adhere to the droplet surface in the ZINC chamber after activation in the IMCA chamber. Therefore, in CLINCH and IMCA/ZINC experiments, the efficiency of contact freezing inside-out is probed and it could be expected that freezing efficiencies in both experiments should be the same. However, the IMCA/ZINC freezing experiments performed with the same AgI aerosol and similar residence times shows a much higher freezing efficiency than the CLINCH experiments. This is in contrast to DeMott (1995), who reported higher freezing efficiencies in contact than in immersion mode for AgI-AgCl aerosols. However, AgI is a complex ice nucleus that appears in different polymorphic forms. Moreover, it partly dissolves in water. Depending on the production procedure, AgI is agglomerated with soluble salts.



Moreover the freezing ability depends on the surface charge on AgI particles. A closer investigation of factors influencing the efficiency of AgI as an ice nucleus is given in the companion paper by Marcolli et al. (2016).

## 5.5 Arizona test dust

The ice nucleation ability of ATD has been investigated by several groups using different setups. Niehaus et al. (2014) investigated contact freezing of deposited droplets on a glass slide, which were exposed to a flow of a polydisperse ATD aerosol (0.3 - 10 μm diameter particles). They determined that one in 1000 particles induced freezing at 253 K, and one in 100 000 at 258 K. These numbers are not directly comparable with this study, because the detection limit for frozen fractions of the IODE detector is ca. 0.05. In CLINCH we observed the onset of freezing at 247 K for 800 nm ATD particles. For 800 nm ATD particles, the freezing efficiencies in contact mode are within experimental uncertainties the same as freezing efficiencies in immersion mode measured with IMCA/ZINC at 3 s residence time. When the ATD parameterization for immersion freezing proposed by Marcolli et al. (2007) is applied to 800 nm particles with a nucleation time of 3 s, it is agreeing well with the experimental data from IMCA/ZINC. Niedermeier et al. (2010) investigated immersion freezing of ATD particles with LACIS. Experiments with ATD aerosols with diameters < 560 nm yielded frozen fractions of 0.04 at 239 K and 0.1 at 236 K. The low residence time and the cutoff of particles > 560 nm might be reasons for this lower freezing efficiency compared with IMCA/ZINC. When the active site parameterization by Marcolli et al. (2007) is applied to 400 nm particles, it gave too high active fractions compared to experiments. The heterogeneous mineralogical composition of ATD may be one of the reasons that smaller particles do not act as effective INPs and may even be inactive. Atkinson et al. (2013) have shown that ATD is composed of 20.3% K-feldspar, 12.4 % (Na, Ca)-feldspar, 17.1 % quartz, 7.5 % illite/muscovite, and 10 % illite/smectite. Clay mineral particles of illite/muscovite tend to be small and presumably dominate the particle fraction with diameters < 500 nm, while quartz and K-feldspar may be overrepresented in the fraction with diameters > 500 nm. Moreover, larger particles are often conglomerates of different minerals (Reid et al., 2003; Kandler et al., 2011) and might contain contributions of some K-feldspar while small particles are often primary particles of one mineral, which might not be very active as INP. Comparison of all measurements shows that immersion and contact freezing are similarly efficient modes of ice nucleation with ATD. Contact freezing experiments performed at 2 s residence time yielded higher freezing efficiencies than at 4 s, which is compatible with contact freezing inside-out or immersion freezing, but not with a collisional freezing mechanism. If particles became immediately immersed after contacting the droplet, freezing would occur in immersion mode also when a contact freezing experiment is performed. Indeed, the surfaces of many mineral dusts like quartz and feldspars are covered with hydroxyl groups, which render surfaces hydrophilic (Koretsky et al., 1997). Shang et al. (2010) measured contact angles of water droplets on clay films and found for illites a dependence of contact angles on relative humidity and on the exchangeable cations: the contact angle of Ca-illite sank from 28.3° to 21.6° when RH was raised from 19 % to 100 %. At 33 % RH contact angles ranged between 23.3° − 34.2° for illites saturated with different cations (Na, K, Mg, or Ca). Contact angles of 31° − 35° were measured for quartz (Szyszka, 2012). When we sprinkled ATD on a water surface, most particles immediately immersed and sank to the bottom. This suggests that when ATD particles collide with water droplets, the particles become immediately immersed such that in immersion freezing and contact freezing experiments the immersion mode is probed.





## 5.6 Kaolinite

X-ray powder diffraction showed that Fluka kaolinite (K-SA) contains only 82.7 % kaolinite, but 5.4 % illite/muscovite, 5.9 % quartz, and 4.5 % K-feldspar (Atkinson et al., 2013). The clay minerals illite/muscovite and kaolinite tend to form small crystals and are presumably enriched in the particle fraction with diameters < 500 nm, while quartz and K-feldspars might be

overrepresented in the fraction with diameters > 500 nm. K-feldspars and illite are known to be efficient ice nuclei (Atkinson et al., 2013; Hiranuma et al., 2015) and may dominate freezing when many particles are present in a sample (Pinti et al., 2012). When only one particle is present, this is likely to be a kaolinite particle. Kaolinite is a clay mineral with the formula $Al_2Si_2O_5(OH)_4$. It has a layered structure with octahedral aluminum and tetrahedral silicon layers. It forms plate-like crystals with sizes of several hundred nanometers to micrometers and typical thicknesses of 30 - 50 nm (Hu and Michaelides, 2007,

2008). These plates have a hydrophilic octahedral Al-OH surface and a rather hydrophobic tetrahedral siloxane (Si-O) surface. The edges of the plates are terminated by oxygen atoms or hydroxyl groups and are hydrophilic. Šolc et al. (2011) computed a contact angle of $105°$ for nanodroplets on the tetrahedral siloxane surface by force-field molecular dynamics. Nanodroplets spread on the octahedral surface indicated a contact angle of $0°$. Shang et al. (2010) measured a contact angle of about $18°$ for water droplets on kaolinite films. This experimental value represents an averaged value over all kaolinite surfaces. The

energetically most favorable configuration is therefore when the kaolinite particle adheres to the water surface with the siloxane surface exposed to air. Whether a kaolinite particle realizes this configuration may depend on the orientation of the particle when it contacts the water droplet. When we sprinkled kaolinite powder on water, we observed that some particles floated on the surface while others became totally immersed and sank to the bottom. The lower freezing efficiency observed for 2 s residence time in the CLINCH chamber compared with 4 s is incompatible with a collisional freezing process but in accordance with

contact freezing inside-out or immersion freezing. A particle on the surface can induce ice nucleation in the immersion mode with the part immersed in water or in contact mode with the part exposed to air. While it is likely that a kaolinite particle that hits a water droplet adheres to the surface and exposes the hydrophobic siloxane surface to air, it is less clear whether particles that underwent droplet activation stick to the surface or whether they totally immerse into the growing droplet. Which is the case may also depend on the conditions during activation like supersaturation or growth rate of the droplet. The immersion and

contact freezing studies compiled in Fig. 6 suggest that contact freezing is more efficient than immersion freezing with an onset temperature that is about 3 K higher. Ladino et al. (2011) who compared their contact freezing data with immersion freezing measurements from Lüönd et al. (2010) using IMCA/ZINC concluded that there are some hints for contact freezing to be more efficient than immersion freezing. Hoffmann et al. (2013b) found that contact freezing dominates over immersion freezing for droplets levitated in an electrodynamic balance that were exposed to a flow of a kaolinite KGa-1b particles. Svensson et al.

(2009) investigated contact freezing using an eletrodynamic balance to levitate droplets exposed to a flow of Fluka kaolinite particles. They observed contact freezing below 249 K for dry conditions and a freezing threshold of 267 K when the air was humidified. This value is higher than the one reported for freezing of bulk suspensions of Fluka kaolinite (K-SA) by Pinti et al. (2012).



It is not clear which surface of kaolinite is responsible for ice nucleation. Using grand canonical Monte Carlo simulations, Croteau et al. (2008, 2010) showed that the Si-O surface remained dry up to water vapor saturation, while the edges and the Al-OH surface are much more hydrophilic and absorb up to a monolayer water at water saturation. Adsorbed water on the octahedral Al-OH surface exhibits hexagonal patterns but no close lattice match with ice (Croteau et al., 2008, 2010).

Simulations by Zielke et al. (2015) showed that for the Al-surface, reorientation of the surface hydroxyl groups is essential for ice nucleation. On the siloxane surface, ice nucleates via an ordered arrangement of hexagonal and cubic ice layers, joined at their basal planes where the interfacial energy cost is low. Experimentally, much higher absorption was determined showing that most absorption probably occurs on surface irregularities such as adsorbed ions or surface defects like trenches, pits, and steps (Schuttlefield et al., 2007; Tabrizy et al., 2011). Croteau et al. (2010) have shown that absorption is much higher on

trenches than on the defect-free surface. Ice nucleation may therefore occur on liquid patches on an otherwise dry surface (Conrad et al., 2005). The wetting state of a nucleus may therefore be a crucial parameter for ice nucleation by kaolinite. This would be in accordance with the higher nucleation temperatures observed by Svensson et al. (2009) at humid conditions.

## 6   Summary and conclusions

This study confirms the findings of Nagare et al. (2015) that theoretical formulations give too low collision efficiencies at sub-

zero temperature for particles in the accumulation mode. In CLINCH, droplets are evaporating giving rise to diffusiophoresis and thermophoresis. Moreover, droplets and particles are charged. Freezing efficiencies calculated from theoretical formulations of collision efficiencies are more than one order of magnitude higher than the highest possible value of $FE = 1$. An assessment of the relevance of contact compared to immersion freezing is therefore limited by knowledge of collision efficiencies. To improve calculated collision efficiencies, formulations of thermophoresis and diffusiophoresis should be re-assessed.

Comparing contact freezing efficiencies acquired at 2 s ($FE(2s)$) and 4 s ($FE(4s)$) residence times enables conclusions regarding the freezing mechanism. For contact freezing experiments with AgI, freezing efficiencies at 2 s and 4 s residence times were the same within error when the droplets collected on average only one particle during their time in the chamber. This is in accordance with a collisional contact freezing mechanism. However, contact freezing inside-out and immersion freezing cannot be excluded if these processes occur at a high rate. For experiments with ATD and kaolinite $FE(2s)$ was

smaller than $FE(4s)$ which is incompatible with immediate freezing after contact. Therefore, immersion freezing or contact freezing inside-out must be at work for these INPs. The comparison of contact and immersion freezing experiments did not confirm a general enhancement of freezing efficiency in contact mode relative to immersion mode. One reason for this may be that in CLINCH and IMCA/ZINC experiments the particles are free to realize the energetically most favorable position in or on the droplet. For AgI particles the freezing efficiency in CLINCH experiments was less than in IMCA/ZINC and the

onset temperature was shifted to lower values. This will be further investigated in Marcolli et al. (2016). For ATD, freezing efficiencies in contact and immersion mode were similar. For kaolinite particles, contact freezing became detectable at higher temperature than immersion freezing. A specific dependence on the INP for the enhancement of contact freezing relative to immersion freezing is in accordance with Gurganus et al. (2014) who observed an increased efficiency for nucleation at the




three-phase contact line in case of nanoscale but not for microscale textures. In most experiments of contact freezing inside-out, the position of the particle with respect to the droplet is fixed by the design of the experiment (Shaw et al., 2005; Fornea et al., 2009; Gurganus et al., 2014). Whether a particle adheres to the surface or becomes totally immersed in a droplet depends on the wetting of the particle with water. A contact angle of zero corresponds with complete wetting, for higher values, the

5  wetting is only partial. Own observations of particles that were gently sprinkled on water confirmed the predictions based on contact angles. Experiments and calculations suggest that AgI particles partition to the droplet surface for contact and immersion freezing experiments. ATD particles seem to have highly hydrophilic surfaces that lead to fast immersion of the particles so that there is no time for contact freezing inside-out and immersion freezing prevails. Kaolinite forms plate-like crystals with a hydrophobic siloxane surface, all other surfaces are hydrophilic. It is therefore energetically most favorable

10  when the hydrophobic surface of kaolinite particles is exposed to air. For this configuration, contact freezing inside-out and immersion freezing can compete.

**Author contribution**

B. Nagare carried out the CLINCH experiments and evaluations. A. Welti carried out IMCA/ZINC experiments. B. Nagare and C. Marcolli prepared the manuscript. O. Stetzer supervised the laboratory work and U. Lohmann supervised the work overall.

15  **Acknowledgments**

This work was supported by the Swiss National Foundation, project 200020_150169. B. Nagare acknowleges the financial support by ETH Zurich. We thank Zamin Kanji, Ulrich Krieger, Jan Henneberger and Joel Corbin for useful discussions.



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



**Table 1.** Instrument characteristics

| Instrument | Droplet diameter (µm) | Residence time of droplet (s) | Aerosol concentration $cm^{-3}$ |
| --- | --- | --- | --- |
| IMCA/ZINC | 18 - 20 | 3 (variable) | 1 particle per droplet |
| CLINCH | $80 \pm 3$ | 2 and 4 | 500 - 5000 |





**Table 2.** Average number of collisions for 800 nm particles with a 80 μm diameter droplet in a concentration of 1000 cm$^{-3}$ and residence times of 2 s and 4 s assuming different values for CE.

| Collision efficiency $CE$ | Number of collisions N | |
| --- | --- | --- |
| | 2 s | 4 s |
| 0.003 | 0.0056 | 0.012 |
| 0.46 | 0.079 | 0.168 |
| 0.13 | 0.24 | 0.48 |





**Table 3.** Average number of collisions $N$ for 200 nm silver iodide particles with a 80 μm diameter droplet in concentrations from 500 to 5000 cm$^{-3}$ and residence times of 2 s and 4 s.

| Concentration cm$^{-3}$ | 2 s | 4 s |
|:---:|:---:|:---:|
| 500 | 0.11 | 0.23 |
| 1000 | 0.23 | 0.47 |
| 2000 | 0.47 | 0.94 |
| 5000 | 1.17 | 2.35 |



**Table A1.** List of symbols

| | | | |
|---|---|---|---|
| $A$ | surface area of the particle [m$^2$] | | |
| $C$ | Concentration of particle [m$^{-3}$] | $CE$ | Collision efficiency |
| $FE$ | Freezing efficiency | $FF$ | Frozen fraction of droplets |
| $L$ | effective length experienced by the droplet [m] | $l$ | length of chamber [m] |
| $R$ | Radius of the droplet [m] | $r$ | Radius of the particle [m] |
| $U(R)$ | teminal velocity of droplet [ms$^{-1}$] | $V_{flow}$ | flow velocity in the chamber [ms$^{-1}$] |
| $\alpha$ | contact angle | $\sigma_{SA}$ | surface tension between particle and air [Jm$^{-2}$] |
| $\sigma_{LA}$ | surface tesnsion between air and liquid [Jm$^{-2}$] | $\sigma_{SL}$ | interfacial tension between particle and water [Jm$^{-2}$] |





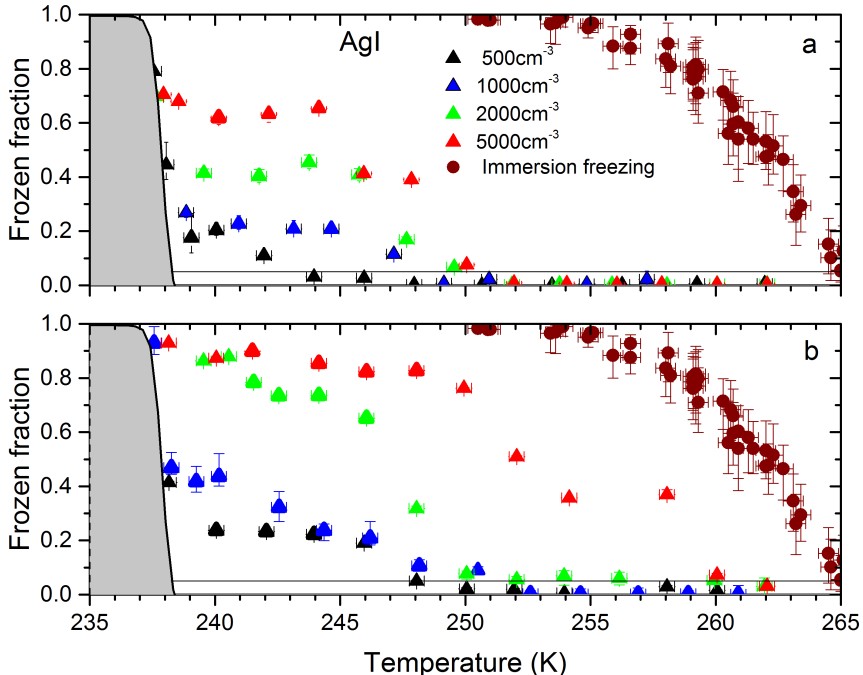

**Figure 1.** Frozen fraction against chamber temperature for silver iodide particles of 200 nm diameter. Contact freezing for aerosol concentrations from $500\,\mathrm{cm}^{-3}$ to $5000\,\mathrm{cm}^{-3}$ are given by triangles for a droplet residence time of 2 s in panel (a) and 4 s in panel (b). Immersion freezing for 3 s residence time of droplets in the ZINC chamber is shown by circles. The gray shaded area shows the homogeneous freezing of droplets determined from blank experiments (without aerosol) and the black horizontal line indicates the lower reliability limits of measurements determined from the blank signal level observed in experiments without aerosol. Error bars in a concentration of 1000 cm-3 and residence times of 2 s and 4 s assuming different values for CE. represent the uncertainty in the frozen fraction due to the classification (liquid or ice) uncertainty of the IODE detector (Lüönd et al., 2010).



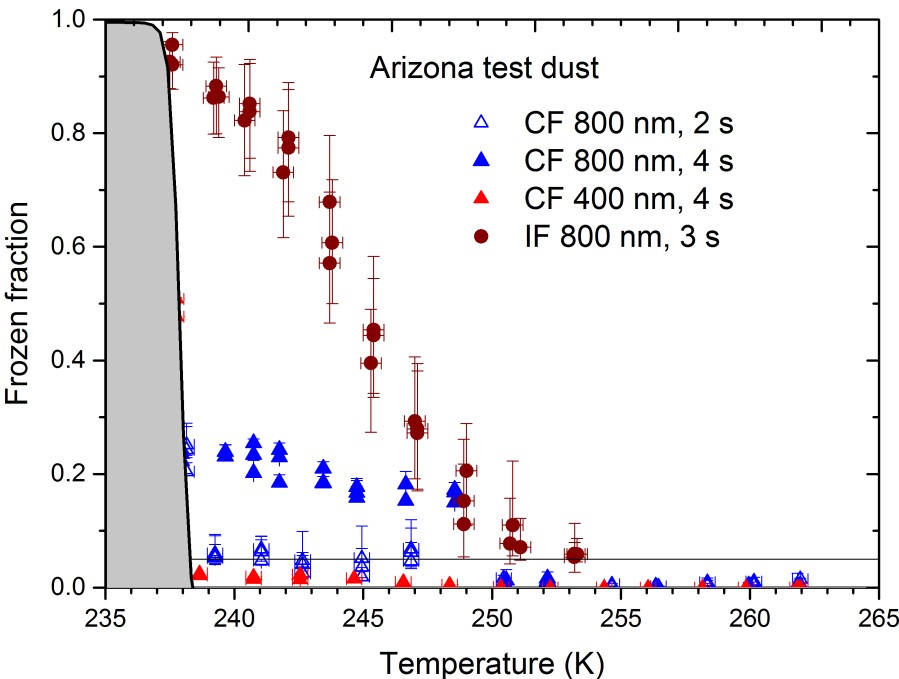

**Figure 2.** Frozen fraction against chamber temperature for ATD particles of 400 nm and 800 nm diameter. Contact freezing (CF) for aerosol concentrations of 1000 $cm^{-3}$ are given by triangles for droplet residence times of 2 s and 4 s. Immersion freezing (IF from IMCA/ZINC) for 800 nm ATD particles and 3 s residence time in ZINC are shown by circles. The gray shaded area shows the homogeneous freezing of droplets determined from blank experiments (without aerosol) and the black horizontal line indicates the lower reliability of the measurements determined from the blank signal level observed in experiments without aerosol. Error bars represent the uncertainty in the frozen fraction due to the classification (liquid or ice) uncertainty of the IODE detector (Lüönd et al., 2010).





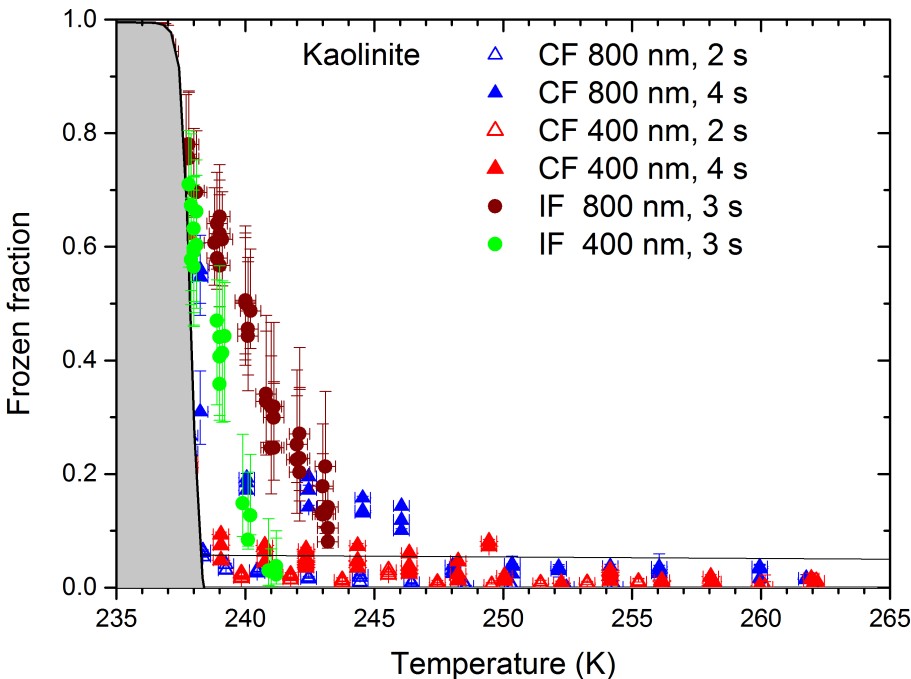

**Figure 3.** Frozen fraction against chamber temperature for kaolinite particles of 400 nm and 800 nm diameter. Contact freezing (CF) for aerosol concentrations of 1000 $\mathrm{cm}^{-3}$ are given by triangles for droplet residence times of 2 s and 4 s. Immersion freezing (IF from IMCA/ZINC) for 400 nm and 800 nm kaolinite particles and 3 s residence time in ZINC are shown by green and brown circles, respectively. The gray shaded area shows the homogeneous freezing of droplets determined from blank experiments (without aerosol) and the black horizontal line indicates the lower reliability of the measurements determined from the blank signal level observed in experiments without aerosol. Error bars represent the uncertainty in the frozen fraction due to the classification (liquid or ice) uncertainty of the IODE detector (Lüönd et al., 2010).





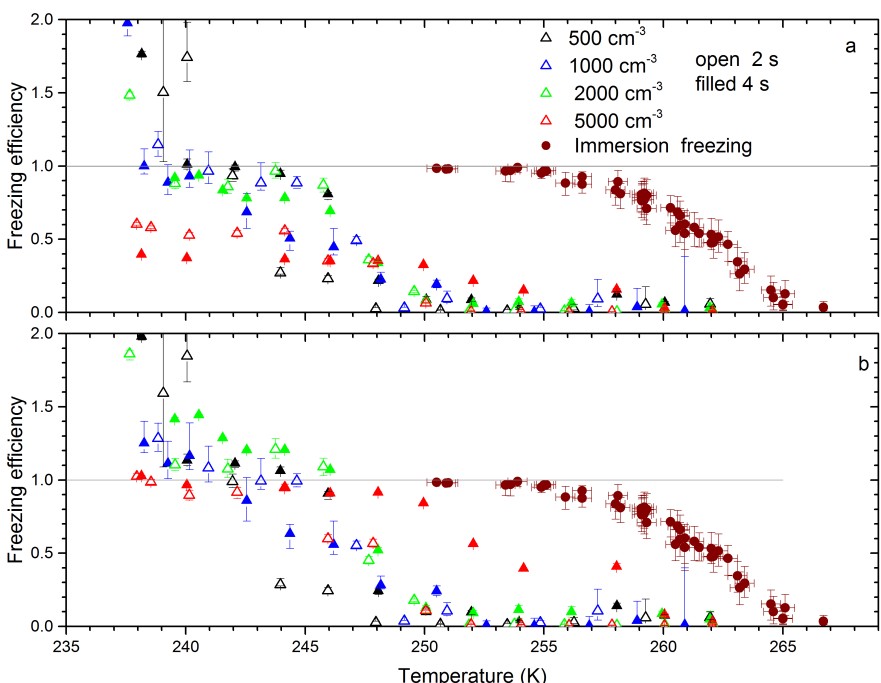

**Figure 4.** Freezing efficiency $FE$ against chamber temperature for contact freezing experiments (triangles) with 200 nm diameter AgI particles with droplet residence times of 2 s (open symbols) and 4 s (filled symbols). The concentration of silver iodide partiles varies from $500\,\mathrm{cm^{-3}}$ to $5000\,\mathrm{cm^{-3}}$. A collision efficiency $CE = 0.13$ is used to calculate $N$. $FE$ is calculated using Eq. (1) in panel (a) and Eq. (4) in panel (b). Immersion freezing of droplets in the ZINC chamber for 3 s residence time is shown as circles. The gray horizontal line indicates the maximum freezing efficiency realized when the first collision initiates freezing.



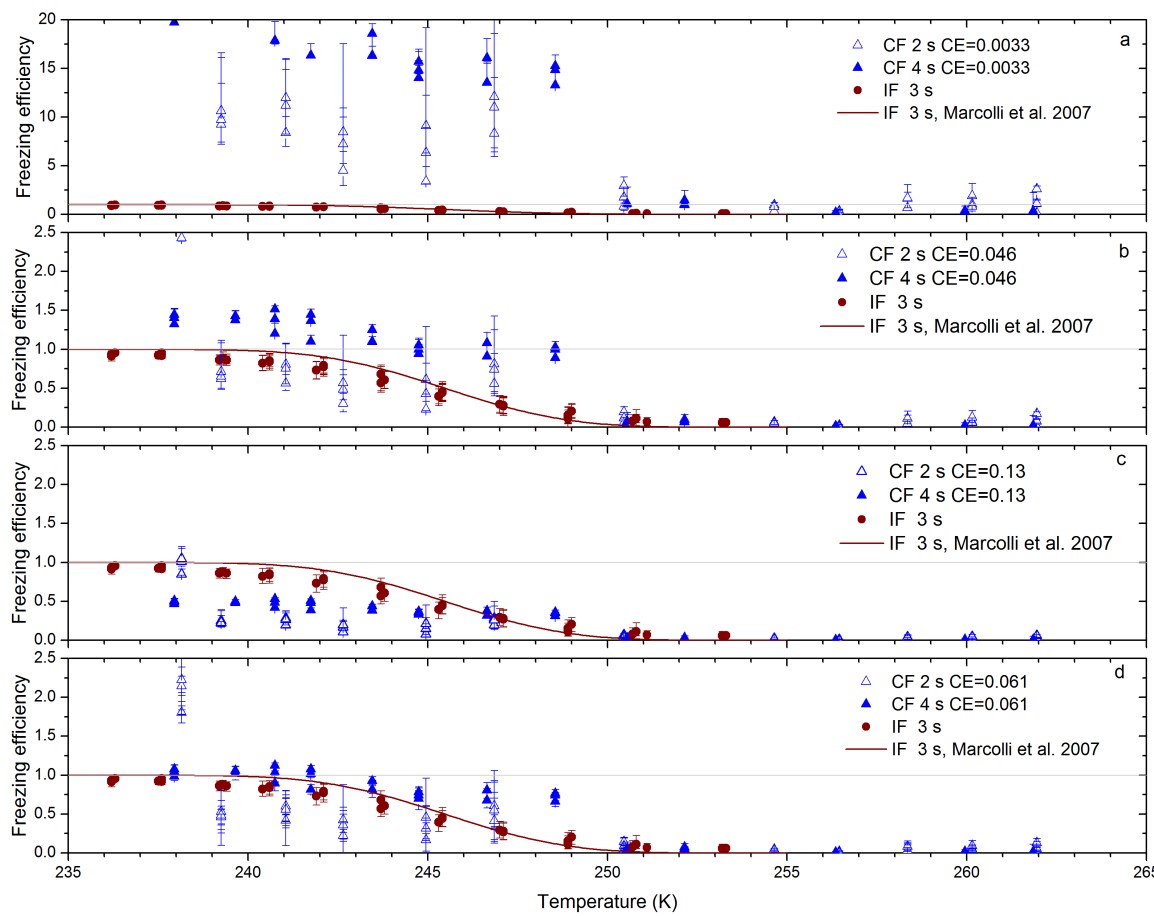

**Figure 5.** Freezing efficiency for 800 nm ATD particles with a concentration of 1000 cm$^{-3}$ calculated with 4 different assumptions for $CE$: panel (a) with theoretical $CE$ from Wang et al. (1978) and Park et al. (2005); panel (b) with $14 \times CE$ from panel (a) (see text for details); panel (c) with $CE = 0.13$ (applying the value for 200 nm AgI particles for all particle sizes); panel (d) with $CE = 0.061$, shifting collision efficiencies close to 1. Filled triangles show contact freezing for 4 s residence time in the CLINCH chamber, the open triangles for 2 s residence time. Each triangle represents an independent measurement. Error bars represent the precision of the IODE detector. Brown circles show the freezing efficiency for immersion freezing of droplets in the IMCA/ZINC chamber for 3 s residence time. The brown lines show the $FF$ calculated with the active site immersion freezing parameterization from Marcolli et al. (2007) evaluated for 800 nm particles and 3 s residence time. The gray horizontal line indicates $FE = 1$. Note that the y-scale in panel (a) is different from the ones in panels (b), (c) and (d).





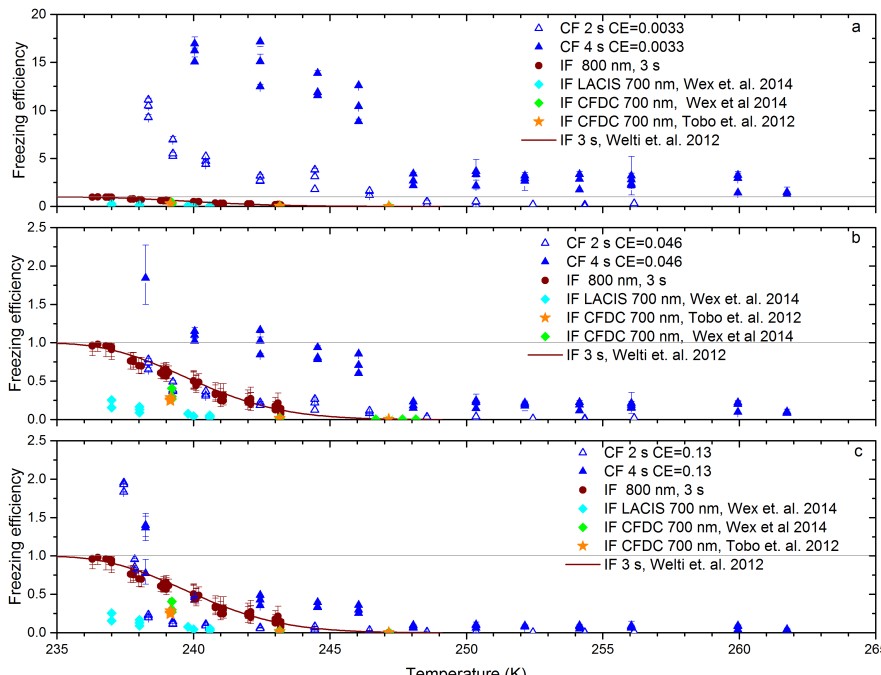

**Figure 6.** Freezing efficiency for 800 nm kaolinite particles with a concentration of 1000 cm$^{-3}$ calculated with 3 different assumptions for $CE$: panel (a) with the theoretical $CE$ from Park et al. (2005) and Wang et al. (1978); panel (b): $14 \times CE$ from panel (a); panel (c) with $CE = 0.13$ (applying value of 200 nm AgI particles) for lower bound of freezing efficiency. Each triangle represents an independent measurement. Error bars represent the precision of the IODE detector. Brown circles show the freezing efficiency of droplets in ZINC for a residence time of 3 s for 800 nm kaolinite particles. Orange stars show immersion freezing of droplets activated by 700 nm Fluka kaolinite particles (from Fig. 2a of Tobo et al. (2012)). Light blue diamonds show immersion freezing of droplets activated by a 700 nm Fluka kaolinite particle (LACIS data with 1.6 s residence time (from Fig. 2 (right panel) of Wex et al., 2014). The green diamond shows immersion freezing of droplets activated by 700 nm kaolinite particles (CFDC data with 5 s residence time (from Fig. 2 (right panel) of Wex et al. (2014)). The brown line represents the $\alpha$-pdf parameterization from Welti et al. (2012). The gray horizontal line indicates the maximum freezing efficiency. Note that the y-scale for panel (a) is different from the ones shown in panels (b) and (c).