# Peer review of "Comparing contact and immersion freezing from continuous flow diffusion chambers"

_Atmospheric Chemistry and Physics, 2016_

## Referee Comment (RC1) · Anonymous Referee #1 · 6 Mar 2016

The manuscript by Nagare et al. describes a comparison between immersion and contact freezing in the ETH Collision Ice Nucleation Chamber and the Immersion Mode Cooling Chamber/Zurich Ice Nucleation Chamber. Unlike previous studies, an enhancement of contact mode over the immersion mode was not seen.

I am supportive of publication. That said, there is some ambiguity in the way that the freezing efficiency is defined and/or used in the paper that should be clarified. In addition, I think that the authors should discuss the discrepancy between their measurements and those of other groups in more detail.

**Freezing efficiency**

I think I am not quite following how $FE$ is defined and used. In the text $FE$ is defined as

$$FE = \frac{FF}{N} \qquad (1)$$

where $FF$ is the fraction of droplets that have frozen and $N$ is defined as the number of collisions between a droplet and aerosol particles. If I combine the definition of $FF$ and equation 1, the result is

$$FE = \frac{\text{freezing events}}{\text{total droplet-aerosol collisions}} \qquad (2)$$

which can be interpreted as the probability that a single collision between an unfrozen droplet and an aerosol particle results in a freezing event. This is the interpretation, used for example, in Hoffmann et al. [1] and Niehaus et al. [2].

My confusion is from the discussion around the case when the first collision between aerosol and droplet results in a freezing event. If that is the case, then intuitively, $FE = 1$. I don't understand the point of Equation 4 in the manuscript. The denominator is defined as the fraction of unfrozen droplets after $N$ collisions, but $1 - e^{-N}$ is defined in Nagare et al. (2015) [3] as $FF$ (see Equation 19). If the denominator *is* $FF$, then Equation 4 in the manuscript makes sense because it reduces to $FE = 1$.

I am also confused by the statement in lines 6 and 7 on page 6. "...$FE$ should be independent of residence time when..." Shouldn't $FE$ always be independent of residence time? The number of freezing events will depend on the residence time, since a longer residence time is a greater probability that the droplet will collide enough particles to catalyze freezing, but once you have normalized by the number of droplet-aerosol collisions, $FE$ should be a probability, which is independent of time.

Figure 4 is also a source of confusion for me. How is $FE > 1$ possible, as is shown in panel b for $T < 245$. If I understand the definition of $FE$ correctly, this implies that there are more freezing events than there are aerosol-droplet collisions. Under the assumption of panel b, shouldn't all these data points collapse to 1 for $T \le 245$? I am also not understanding how $FE$ is a function of the aerosol concentration. $FF$ *should* be a function of the concentration, but the point of normalizing by the number of collisions is to remove that dependence.

**Immersion vs. contact freezing**

Based on a difference in the freezing efficiency with a residence time of 2 seconds vs. 4 seconds, the authors conclude that contact freezing is not enhanced over immersion freezing, at least not for the Arizona Test Dust, and perhaps not for silver iodide. This claim is also supported by the comparison to immersion freezing results using IMCA/ZINC. (See my discussion above concerning confusion about $FE$ and a time dependence.)

This is a striking result, considering the previous work in the field. Gokhale and Goold [4] tested AgI for contact freezing, and showed that freezing was initiated at temperatures close to -5 °C in the contact mode while temperatures closer to -15 °C were necessary for immersion mode freezing. Sax and Goldsmith [5] also tested AgI, in a cold room with freely suspended droplets, and saw indications of a shift to lower temperatures for the onset of freezing for contact vs. immersion freezing. These results are directly relevant to the experiments described here. While the freezing efficiency is not reported in either of these papers, they do discuss a shift in the onset freezing temperature, which can be compared, for example, to Figure 1 in this paper, which unambiguously shows that droplets begin to freeze at about -8 °C in the immersion mode and only at -13 °C in the $FE(4s)$ experiments in the contact mode. (See also Gokhale and Spengler [6] and Gokhale and Lewinter [7]. Tests with AgI were conducted in both of those studies as well.)

Similarly, Pitter and Pruppacher [8] tested kaolinite in a wind tunnel and found a clear shift to lower temperatures for the onset of freezing when changing from contact to immersion mode experiments. Niehaus et al. [9] report that they ran experiments in which a freezing event occurred, then melted the droplet and cooled it back to the original temperature. No freezing events occurred. Niehaus et al. concluded that these tests, in which the same aerosol were compared against themselves, that contact freezing was more probable than was immersion mode. (ATD was among the substances they tested.) Though AgI, kaolinite, nor ATD was tested in the study, in an even more convincing case of an enhancement of contact freezing over immersion mode, Durant and Shaw [10], visually confirmed a shift in the freezing temperature when a particle was at the air-water interface. (See their Figure 1.)

As noted above, I am in favor of publication because I believe that more information on these systems will help the community to unravel the complexities and, perhaps, make a determination as to just how important contact freezing is in Earth's atmosphere. That said, I believe that the authors should place their results more clearly in the context of previous work.

**Other comments**

**Discussion of uncertainties:**The authors discuss the uncertainty in the frozen fraction as stemming from the classification uncertainty of the IODE detector (see, e.g. the caption to Fig. 5.) Shouldn't the uncertainty in the number of

particles that have collided with the droplet be included when showing $FE$? I realize that different values of $N$ are used, depending on the value of $CE$, but even so, there is an uncertainty in $CE$ and thus $N$. An indication of that uncertainty would help in interpreting these plots.

**Discussion of contact freezing mechanisms:** Pg. 8, line 29. "This mechanism was refuted by Fukuta (1975b)." "Refute" implies that he disproved Cooper's mechanism. I think he rejected it, but did not disprove it.

Pg. 9, line 18. "This indicates that collision itself does not increase $FE$..." There is evidence that the collision can increase $FE$. Davis et al. [11] recently showed that salt particles can initiate efflorescence upon contact with a supersaturated solution of a different salt. Niehaus and Cantrell [12] observed freezing events initiated by soluble salts at temperatures above the eutectic. Finally, Yang et al. [13] have observed freezing which may be in response to the movement of the triple line, as from a collision.

pg. 13, line3 28-29: "One reason for this may be that in CLINCH and IMCA/ZINC experiments the particles are free to realize the energetically most favorable position in or on the droplet." This may be true, but it is also true for many of the other experiments noted above.

pg. 14, line 5: "Own observations..." I think you mean to start that sentence with "Our".

Table A1: Include $N$.

**Bibliography**

[1] N. Hoffmann, D. Duft, A. Kiselev, and T. Leisner. Contact freezing efficiency of mineral dust aerosols studied in an electrodynamic balance: quantitative size and temperature dependence for illite particles. *Faraday Discuss.*, 165:383–390, 2013.

[2] J. Niehaus, K.W. Bunker, S. China, A. Kostinski, C. Mazzoleni, and W. Cantrell. A technique to measure ice nuclei in the contact mode. *J. Atmos. Ocean. Technol.*, 71:3659–3667, 2014.

[3] B. Nagare, C. Marcolli, O. Stetzer, and U. Lohmann. Comparison of measured and calculated collision efficiencies at low temperatures. *Atmos. Chem. Phys.*, 15(23):13759–13776, 2015.

[4] N. Gokhale and J. Goold, Jr. Droplet freezing by surface nucleation. *J. Appl. Meteorol.*, 7:870–874, 1968.

[5] R. Sax and P. Goldsmith. Nucleation of water drops by brownian contact with AgI and other aerosols. *Q. J. Roy. Meteorol. Soc.*, 98:60–72, 1972.

[6] N. Gokhale and J. Spengler. Freezing of freely suspended, supercooled water drops by contact nucleation. *J. Appl. Meteorol.*, 11:157–160, 1972.

[7] N. Gokhale and O. Lewinter. Microcinematographic studies of contact nucleation. *J. Appl. Meteorol.*, 10:469–473, 1971.

[8] R. Pitter and H. Pruppacher. A wind tunnel investigation of freezing of small water drops falling at terminal velocity in air. *Q. J. Roy. Meteorol. Soc.*, 99:540–550, 1973.

[9] J. Niehaus, J. G. Becker, A. Kostinski, and W. Cantrell. Laboratory measurements of contact freezing by dust and bacteria at temperatures of mixed-phase clouds. *J. Atmos. Sci.*, 71(10):3659–3667, 2014.

[10] A. J. Durant and R. A. Shaw. Evaporation freezing by contact nucleation inside-out. *Geophys. Res. Lett.*, 32(20):L20814, 2005.

[11] R. D. Davis, S. Lance, J. A. Gordon, S. B. Ushijima, and M. A. Tolbert. Contact efflorescence as a pathway for crystallization of atmospherically relevant particles. *PNAS*, 112(52):15815–15820, 2015.

[12] J. Niehaus and W. Cantrell. Contact freezing of water by salts. *J. Phys. Chem. Lett.*, 6(17):3490–3495, 2015.

[13] F. Yang, R. A. Shaw, C. W. Gurganus, S. K. Chong, and Y. K. Yap. Ice nucleation at the contact line triggered by transient electrowetting fields. *Applied Physics Letters*, 107(26):264101, 2015.

---

## Referee Comment (RC2) · A. Kiselev (Referee) · 15 Mar 2016

**Reviewer comments to the manuscript "Comparing contact and immersion freezing from continuous flow diffusion chambers" by Baban Nagare, Claudia Marcolli, André Welti, Olaf Stetzer, and Ulrike Lohmann**

*Alexei Kiselev and Nadine Hoffmann (contact: alexei.kiselev@kit.edu)*

*Karlsruhe Institute of Technology, Institut of Meteorology and Climate Research, Karlsruhe, Germany*

The manuscript of Nagare et al. describes contact freezing experiments conducted with supercooled droplets freely falling through the chamber containing ice nucleating (IN) aerosol particles. At the first glance, this seems to be the ideal experimental setup to study the evasive phenomena of contact freezing. The temperature regime, humidity, droplet and particle size are all atmospheric relevant. Already the second glance reveals the difficulties in measuring the magnitudes in question (collision efficiency, CE, and freezing efficiency, FE) and interpreting the observation results. Most puzzling, the authors report no enhancement of contact freezing compared to the immersion freezing conducted with the same IN particles and in similar setup. We share the attitude of the Reviewer 1 that the manuscript should be published to provide the basis for the discussion of the presented measurements and data interpretation. There are some issues, however, that could be improved at the stage of preparation of the final manuscript that we would like to discuss here.

To our opinion, the main conclusion about the role of contact freezing made in this paper is a direct consequence of the method used to measure the CE and the approach used to compare the contact and immersion freezing behavior. We discuss these issues below followed by more specific remarks.

1. The uncertainty in interpretation of contact freezing results is obviously related to the uncertainty of determination of collision efficiency. The CE experimentally determined for 0.2 μm AgI particles (0.13) was reported being 14 times larger than calculated theoretical value. What could be the reason for such a high discrepancy? The degree of control of the experimental parameters (temperature, humidity, size and evaporation rate of the droplets, particle number concentration, droplet charge etc.) is very high, and all known interaction forces seem to be taken into account. On the other hand, the sensitivity of experimental observables to the value of the CE is very strong (see equation 19 and discussion of figure 4 in (Nagare et al., 2015), so that its knowledge is crucial for drawing a conclusion about the role of the contact freezing.

   One of the possible explanations would be the depletion of the IN particle concentration within the volume swept by the droplet train. Water droplets generated with 100 Hz frequency and falling with 0.186 m/s terminal velocity would be separated by 2 mm distance or 0.01 s time lag. The RMS diffusion displacement calculated for 0.2 μm diameter particle (Hinds, 1999, equation 7.18) is about 2e-6 m. Assuming the CE for 0.2 μm AgI particles equal to 0.13, the radius of the cylindrical volume where a falling 80 μm droplet experiences collisions with aerosol particles is
   $$r_{drop} \times (CE)^{0.5} = 4 \times 10^{-5}m \times (0.13)^{0.5} = 1.4 \times 10^{-5}m,$$
   seven times larger than the RMS Brownian displacement of the AgI particle within the droplet interarrival time. This essentially means that the particle number concentration reduced due to the scavenging by falling droplet will not return to equilibrium before the next droplet arrives. Reduced number concentration has to be compensated by higher apparent freezing efficiency (according to equation 19 from (Nagare et al., 2015) to describe the observed fraction of frozen droplets. This back-of-the-envelope calculation shows that the depletion of aerosol in the droplet train zone is quite possible and might affect the calculation of the collision efficiency. For larger aerosol particles the depletion can be even larger.

2. We share the confusion of reviewer 1 with respect to the discussion of freezing efficiency calculated with equation 1 or equation 4 (Section 4.1). In your preceding paper (Nagare et al., 2015) the CE has been calculated with equation 19, which is just equation 4 of this manuscript under assumption that FE = 1. To my understanding, with the CE defined in this way the FE should be derived using equation 4 and not with the equation 1. It is correct that the time independence of FE should indicate the contact freezing but it has nothing to do with the number of collisions required to induce the freezing, it can happen on the first collision or after several dozens of them and still have to be the dominant mechanism. Surely the choice of equation could not be helpful to decide which freezing mechanism is dominating the apparent freezing rate of the droplets.

3. Why is the freezing induced by a particle adhering to the surface of the droplet called "freezing inside-out" throughout the paper? In the original paper (Durant and Shaw, 2005) the IN particle was penetrating the surface from inside of an evaporating droplet, hence the name. In the present manuscript this name is used to describe the situation where an IN particle adheres to the surface and is only partly immersed into the droplet, as compared to the fully immersion freezing mode in IMCA/ZINC. To our understanding, partial immersion does not imply a new nucleation mechanism different. As have been shown in (Hoffmann et al., 2013a), the contact freezing efficiency of mineral dust IN particles is proportional to the surface area of the particle. We argue there that the term "contact freezing" does not imply freezing on a point contact but a considerable fraction of particle surface has to be involved into the freezing process. The term "freezing inside-out" was used to highlight the process of penetration of the droplet surface, and is not fully applicable in the present manuscript. Please consider removing this term from the paper.

4. Based on the comparison of FF and freezing onset temperatures you conclude that the contact freezing is a not dominant freezing process. This conclusion seems questionable, because to our opinion the true value of a contact FE for AgI cannot be derived from the experiment. A better way to compare the two freezing process is based on their characteristic times, as suggested in (Hoffmann et al., 2013b). There we have introduced a characteristic residence time $t_{im}$ of a supercooled droplet experiencing collisions with the IN particles as

$$t_{im} = \frac{2 \cdot FE_{contact}}{J_{imm}},$$

where $J_{imm}$ is the rate of freezing due to immersion freezing and can be estimated from the IMCA/ZINC measurements using the relationship between the number of unfrozen and total number of droplets:

$$\frac{N_{unfrozen}}{N_{total}} = 1 - FF = \exp(-J_{imm} \cdot t)$$

and the residence time in ZINC of $t = 3\ s$. At T = 255K the FF for the AgI 0.2 μm particles in the immersion freezing experiment (we refer to figure 1) is ≈0.95 for number concentration of 5000 cm$^{-3}$ and therefore $J_{imm} = 1s^{-1}$. The meaning of this is that on average, droplets would freeze in immersion mode 1 second after collision with 200 μm AgI particle at this temperature. Even if we assume the $FE_{contact} = 1$ (in the figure 4 it is rather 0.3 to 0.5) the characteristic time would be $t_{im} \approx 2s$, comparable to the shortest residence time used in your experiment, and thus the condition

$$FE_{contact} \gg \frac{1}{2}J_{imm}t$$

required to observe the dominance of contact freezing is not fulfilled. This simple analysis show that at least for silver iodide particles both freezing mechanisms are competing and there is no way to derive the FE for both mechanisms separately on the time scale of the experiment.

As it is immediately follows from these considerations, the freezing on-set in purely immersion mode should be always observed at higher temperature just due to the fact, that in INCA/ZINC droplets are entering the cold zone carrying the IN particles inside, whereas time is needed in CLINCH for droplet first to collect an IN particle and then freeze due to one of the freezing mechanisms.

5. Difference or equality of FE at different reference times is discussed throughout the manuscript. However, the difference in FE for different aerosol concentrations (as seen in the figure 4) is neglected. This behavior cannot be explained by interplay of the immersion vs. contact freezing as it is done for the residence time dependence.

**Specific comments**

1. There is an apparent contradiction between two statements (page 6, lines 21-25).: "*Panel (a) of Fig. 4 shows that FE does not exceed 0.5 for C =5000 cm−3 because it is assumed that on average 2.35 collisions are necessary to freeze a droplet. This led us to Eq. (4) to calculate FE, which assumes that already the first collision induces droplet freezing*". If more than one collision is needed to freeze the droplet, the freezing could occur on any of the subsequent collisions, couldn't it?

2. (page 6, lines 25-26).: "*This reinforces the assumption that the first contact leads to droplet freezing in this temperature range and confirms the plateau condition used in Nagare et al. (2015) to derive CE.*" This is a confusing statement: in your previous paper the CE was derived from the measurements of FF under assumption that FE = 1 (in the plateau region, at T < 245K), and now you derive the FE value from essentially the same measurements under assumption of known CE? In this case you can't obtain any other value of FE but 1.

3. (page 10, line 28) Could you clarify why is the solubility of silver iodide important for the contact freezing experiments reported in this paper and what do mean by the statement "*Moreover the freezing ability depends on the surface charge on AgI particles*" (page 11 line 1).

**References**

Durant, A. J. and Shaw, R. A.: Evaporation freezing by contact nucleation inside-out, Geophys. Res. Lett., 32(20), 2–5, doi:10.1029/2005GL024175, 2005.

Hinds, W. C.: Aerosol technology: properties, behavior, and measurement of airborne particles, Wiley Interscience, 1999.

Hoffmann, N., Duft, D., Kiselev, A. and Leisner, T.: Contact freezing efficiency of mineral dust aerosols studied in an electrodynamic balance: quantitative size and temperature dependence for illite particles, Faraday Discuss., 165(0), 383, doi:10.1039/c3fd00033h, 2013a.

Hoffmann, N., Kiselev, A., Rzesanke, D., Duft, D. and Leisner, T.: Experimental quantification of contact freezing in an electrodynamic balance, Atmos. Meas. Tech., 6(9), 2373–2382, doi:10.5194/amt-6-2373-2013, 2013b.

Nagare, B., Marcolli, C., Stetzer, O. and Lohmann, U.: Comparison of measured and calculated collision efficiencies at low temperatures, Atmos. Chem. Phys., 15(23), 13759–13776, doi:10.5194/acp-15-13759-2015, 2015.

---

## Referee Comment (RC3) · Anonymous Referee #3 · 16 Mar 2016

My major comment is that main conclusions of the paper are not visible. Some revision is needed to enhance the readability, and also it is necessary to clarify what are the major conclusions of the paper. It is also not clear why this study is important, and what the atmospheric implications are. Some more discussion is needed to understand why AgI, ATD and Kaolinite particles were used, why natural dust or soil dust particles were not chosen as these are more atmospherically relevant. This is nice study, overall contact freezing is not well understood, but main message is buried. Below some comments may help to revise this paper further.

-What is the typical size of supercooled droplets observed in mixed phase clouds? How often 80 um droplets are observed. Atmospheric relevance of droplet size should be discussed.

[Figure]

-Following two sentences (i and ii) needs to be elaborated. Bulk liquid water properties are different from individual water droplet properties. Please define what you mean by sprinkling. Do particles were size-selected, how many particles were used, what is the temperature of the liquid water, do water is pure or distilled or regular lab supply grade, how long this experiment was performed, do all particles sediment, and how this observation was made (visual observation, microscope).

(i) "When we sprinkled ATD on a water surface, most particles immediately immersed and sank to the bottom. This suggests that when ATD particles collide with water droplets, the particles become immediately immersed such that in immersion freezing and contact freezing experiments the immersion mode is probed."

(ii) "When we sprinkled kaolinite powder on water, we observed that some particles floated on the surface while others became totally immersed and sank to the bottom."

-It is mentioned that "A particle on the surface can induce ice nucleation in the immersion mode with the part immersed in water or in contact mode with the part exposed to air." How this can be assumed, what is the basis for this?

-Section 5.6: It is not clear what results are discussed. This section looks like reading a literature review. There is only one sentence (The immersion and contact freezing studies compiled in Fig. 6 suggest that contact freezing is more efficient than immersion freezing with an onset temperature that is about 3 K higher), which describes the results, but there is no discussion. I suggest use present results to discuss the figure 6, but not previous results (as they have different instrument platform to study Kaolinite properties). For example XRD analysis of Kaolinite particles differ from group to group because of the XRD instrument sensitivity issues, and also impurities within the Kaolinite samples. Note that Kaolinite from different vendors have different properties, also shown by Wex et al (http://www.atmos-chem-phys.net/14/5529/2014/acp-14-5529-2014.pdf) who shown ice nucleating properties are sensitive to the particles procured from different vendors.

**[ACPD](https://www.atmos-chem-phys-discuss.net/)**
-Section 5.6: Second paragraph. How this is applicable to the present study. This material is not relevant, if yes please discuss how. As mentioned above this reads like a literature review.

-Please see Section 5.5 too. Discuss the present results. There is lot of discussion on previous studies, but how they are related to this study. It is not clear why these studies are discussed. I suggest move this material to Intro section to increase the readability.

-Last three sentences from Conclusion section (page 14, line 8-11). Do authors performed any experiments to conclude this, or these are the conclusions from previous studies. If later then I suggest move this to intro section.

-Can majority of Section 5.2 (except page 17, line 17-23) and Section 5.3 be moved to Intro section? They do not discuss any results.

-It may be a good idea to combine section 4 and 5. Section 5, for dust particles, has lot of discussion concerning previous studies and may help to increase the readability.

---

## Author Comment (AC1) · 24 May 2016

The manuscript by Nagare et al. describes a comparison between immersion and contact freezing in the ETH Collision Ice Nucleation Chamber and the Immersion Mode Cooling Chamber/Zurich Ice Nucleation Chamber. Unlike previous studies, an enhancement of contact mode over the immersion mode was not seen.

I am supportive of publication. That said, there is some ambiguity in the way that the freezing efficiency is defined and/or used in the paper that should be clarified. In addition, I think that the authors should discuss the discrepancy between their measurements and those of other groups in more detail.

*We thank the reviewer for the careful reading of the manuscript and the suggestions for improvement. We address the comments below point by point (in italic):*

**Freezing efficiency**

I think I am not quite following how $FE$ is defined and used. In the text $FE$ is defined as

$$FE = \frac{FF}{N} \tag{1}$$

where $FF$ is the fraction of droplets that have frozen and $N$ is defined as the number of collisions between a droplet and aerosol particles. If I combine the definition of $FF$ and equation 1, the result is

$$FE = \frac{freezing\ events}{total\ droplet-aerosol\ collisions} \tag{2}$$

which can be interpreted as the probability that a single collision between an unfrozen droplet and an aerosol particle results in a freezing event. This is the interpretation, used for example, in Hoffmann et al. [1] and Niehaus et al. [2]. My confusion is from the discussion around the case when the first collision between aerosol and droplet results in a freezing event. If that is the case, then intuitively, $FE = 1$. I dont understand the point of Equation 4 in the manuscript. The denominator is defined as the fraction of unfrozen droplets after $N$ collisions, but $1 - e^{-N}$ is defined in Nagare et al. (2015) [3] as $FF$ (see Equation 19). If the denominator *is* $FF$, then Equation 4 in the manuscript makes sense because it reduces to $FE = 1$.

*There is an ambiguity in the freezing efficiency determined by the CLINCH experiment when the droplet collides with more than 1 particle while it passes through the chamber, because in this case, it is not clear which collision is responsible for freezing. A second or a third collision might occur although they are not needed to freeze the droplet when it has frozen already on the first collision. To convert from FF to FE, only collisions of particles with liquid droplets should be counted. The reviewer is right that Eq. 4 should only be used when FE = 1. This is the case for AgI at temperatures for which the plateau condition applied in Nagare et al. (2015) is valid. Using Eq. (1), FE at the highest AgI concentration is lowest. A result that is not easily explained. Therefore, we introduced Eq. (4) for this case.*

*From this it can be concluded that experiments should be run such that the average number of collisions is close to but below one. In this case, Eq. (1) of the manuscript can always be used. We have improved the discussion of Eqs. (1) and (4) in the revised manuscript.*

I am also confused by the statement in lines 6 and 7 on page 6. "...*FE* should be independent of residence time when..." Shouldnt *FE* always be independent of residence time? The number of freezing events will depend on the residence time, since a longer residence time is a greater probability that the droplet will collide enough particles to catalyze freezing, but once you have normalized by the number of droplet-aerosol collisions, *FE* should be a probability, which is independent of time.

*In the case of collisional contact freezing, FE should be indeed independent of time. However, in the case of immersion freezing and adhesion freezing, FE depends on time. This is why we use FE(4s) > FE(2s) as criterion against collisional contact freezing. We have improved the text in the revised manuscript to make this clearer.*

Figure 4 is also a source of confusion for me. How is *FE* > 1 possible, as is shown in panel b for *T* < 245. If I understand the definition of *FE* correctly, this implies that there are more freezing events than there are aerosol-droplet collisions. Under the assumption of panel b, shouldnt all these data points collapse to 1 for *T* ≤ 245? I am also not understanding how *FE* is a function of the aerosol concentration. *FF* should be a function of the concentration, but the point of normalizing by the number of collisions is to remove that dependence.

*FE > 1 is due to homogeneous freezing and measurement uncertainties. We add this statement to the revised manuscript. Indeed, FE should not be a function of concentration.*

**Immersion vs. contact freezing**

Based on a difference in the freezing efficiency with a residence time of 2 seconds vs. 4 seconds, the authors conclude that contact freezing is not enhanced over immersion freezing, at least not for the Arizona Test Dust, and perhaps not for silver iodide. This claim is also supported by the comparison to immersion freezing results using IMCA/ZINC. (See my discussion above concerning confusion about *FE* and a time dependence.)

This is a striking result, considering the previous work in the field. Gokhale and Goold [4] tested AgI for contact freezing, and showed that freezing was initiated at temperatures close to -5 ˚C in the contact mode while temperatures closer to -15 ˚C were necessary for immersion mode freezing. Sax and Goldsmith [5] also tested AgI, in a cold room with freely suspended droplets, and saw indications of a shift to lower temperatures for the onset of freezing for contact vs. immersion freezing. These results are directly relevant to the experiments described here. While the freezing efficiency is not reported in either of these papers, they do discuss a shift in the onset freezing temperature, which can be compared, for example, to Figure 1 in this paper, which unambiguously shows that droplets begin to freeze at about -8 ˚C in the immersion mode and only at -13 ˚C in the *FE*(4*s*) experiments in the contact mode. (See also Gokhale and Spengler [6] and Gokhale and Lewinter [7]. Tests with AgI were conducted in both of those studies as well.)

*We expected to observe a higher freezing efficiency for AgI in CLINCH than in IMCA/ZINC. We were very astonished that we observed the opposite. We went therefore back to literature and studied the previous work. Because so many studies have been performed with AgI as ice nucleus, we decided that a profound discussion of all this literature would make the paper too long. Therefore we wrote a companion paper, which reviews ice nucleation studies with AgI. This paper is now also published in ACPD: Marcolli C., Nagare, B., Welti, A., and Lohmann U.: Ice nucleation efficiency of AgI: review and new insights, Atmos. Chem. Phys. Discuss., doi:10.5194/acp-2016-142, 2016.*

*In this paper, we refer to the literature mentioned by the reviewer. On page 4, we write:*

*Gokhale and Goold (1968) performed contact nucleation experiments by sprinkling AgI particles on supercooled droplets on a hydrophobic plate. They observed that the particles (5 – 400 µm in diameter) remained on the surface of the drops and initiated freezing at the initial stage temperature of 268 K. However, they did not quantify the number of particles present, which precludes an evaluation in terms of surface area. They performed similar experiments for an AgI smoke produced from an AgI string generator with particle diameters from 50 – 100 nm. These particles initiated freezing of 50 % of droplets at 263 K when the stage was cooled at a rate of 1.3 K/min. Gokhale and Goold (1968) concluded that these freezing temperatures are 5 – 10 K higher than the ones observed by Hoffer (1961) for droplets embedded in an oil with immersed AgI particles and attributed it to an enhanced freezing probability for dry particles on a surface compared with particles immersed in the droplet.  However, a strict comparison is not possible because in both studies, information is lacking to quantify the surface area present per droplet. In a follow-up study, Gokhale and Lewinter (1971) monitored the freezing process of 2 mm water droplets with a movie camera and observed that nucleation was initiated at the point of particle contact and continued from there over the entire surface of the drop. The interior of the drop froze at a much slower rate.*

*On page 5, we write:*

*Sax and Goldsmith (1972) performed contact and immersion freezing experiments in a cloud chamber. Freely falling droplets with diameters of 40 – 160 µm (average: 100 µm) intercepted a horizontal aerosol stream of $5 \cdot 10^6$ $cm^{-3}$ AgI particles with 30 nm diameter (size range from 10 – 40 nm) for 0.04 s (1 cm in vertical extent). The aerosol was produced by heating an AgI-coated resistance wire to T = 700°C in a nitrogen stream.  For contact freezing experiments the droplets were brought in thermal equilibrium before intercepting the aerosol stream. After coagulation with the AgI particles, the droplets proceeded into an observation chamber where frozen droplets were*

*distinguished visually from liquid ones. Coagulation of 100 µm droplets with 30 nm particles were dominated by Brownian motion. Assuming a collision efficiency of ca. 0.3, around 100 particles would be captured by the droplet (note that this number is higher than the collection of only 1 particle estimated by Sax and Goldsmith, 1972). For immersion freezing experiments, the droplets passed the aerosol stream at T > 273 K, before they were cooled to the target temperature. Residence time in the chamber was around 4 s. Immersion freezing occurred at 2 K lower temperature than contact freezing.*

*In Marcolli et al. (2016) we give possible explanations why FE in IMCA/ZINC was higher than in CLINCH. To avoid telling the same in two papers, we do not want to extend the discussion in the present manuscript but prefer to refer to the companion paper.*

Similarly, Pitter and Pruppacher [8] tested kaolinite in a wind tunnel and found a clear shift to lower temperatures for the onset of freezing when changing from contact to immersion mode experiments. Niehaus et al. [9] report that they ran experiments in which a freezing event occurred, then melted the droplet and cooled it back to the original temperature. No freezing events occurred. Niehaus et al. concluded that these tests, in which the same aerosol were compared against themselves, that contact freezing was more probable than was immersion mode. (ATD was among the substances they tested.) Though AgI, kaolinite, nor ATD was tested in the study, in an even more convincing case of an enhancement of contact freezing over immersion mode, Durant and Shaw [10], visually confirmed a shift in the freezing temperature when a particle was at the air-water interface. (See their Figure 1.)

*Thank you for pointing out the study of Pitter and Pruppacher. We now refer to it in the revised manuscript. We referred to the Niehaus et al. (2014) paper in the introduction and discussed it in Section 5.5. We add in the revised manuscript that Niehaus et al. concluded that contact freezing was more probable than immersion freezing for the ice nuclei that they investigated.*

As noted above, I am in favor of publication because I believe that more information on these systems will help the community to unravel the complexities and, perhaps, make a determination as to just how important contact freezing is in Earth's atmosphere. That said, I believe that the authors should place their results more clearly in the context of previous work.

**Other comments**

**Discussion of uncertainties:** The authors discuss the uncertainty in the frozen fraction as stemming from the classification uncertainty of the IODE detector (see, e.g. the caption to Fig. 5.). Shouldnt the uncertainty in the number of particles that have collided with the droplet be included when showing $FE$? I realize that different values of $N$ are used, depending on the value of $CE$, but even so, there is an uncertainty in $CE$ and thus $N$. An indication of that uncertainty would help in interpreting these plots.

*We agree that the main uncertainty for the comparison of freezing mechanisms results from uncertainties in CE. We think that it is more transparent to show results with different assumptions of CE rather than to draw huge error bars. To make it clearer that the uncertainty of FE is due to the uncertainty of CE, we added the following sentence to the Section 4.3 in the revised manuscript: "The difference in FE between panels (c) and (d) must be considered as an uncertainty in FE due to the lack of reliable theoretical values of CE in the investigated temperature and particle size range." In Section 4.4 we add: "For panel (a) the theoretical formulations were used, while panels (b) and (c) give the upper and lower limit of FE, respectively."*

**Discussion of contact freezing mechanisms:** Pg. 8, line 29. "This mechanism was refuted by Fukuta (1975b). "Refute" implies that he disproved Cooper's mechanism. I think he rejected it, but did not disprove it.
*Thank you for pointing this out, we change it as suggested.*

Pg. 9, line 18. "This indicates that collision itself does not increase $FE$..." There is evidence that the collision can increase $FE$. Davis et al. [11] recently showed that salt particles can initiate efflorescence upon contact with a supersaturated solution of a different salt. Niehaus and Cantrell [12] observed freezing events initiated by soluble salts at temperatures above the eutectic. Finally, Yang et al. [13] have observed freezing which may be in response to the movement of the triple line, as from a collision.
*Thank you for pointing out these papers. Niehaus and Cantrell indeed made plausible that the collision triggered freezing. To have an effect, the particles that collided with the droplets needed to be large so that the impact led to a mechanical disturbance. For 10 µm NaCl particles no effect was observed. Therefore, this process is likely not active in our experiment. However, we mention this study in the revised manuscript to illustrate that the collision may induce freezing when the impact is large enough. Davis et al. investigated contact efflorescence by bringing soluble salts in contact with supersaturated solution droplets. In this case, nucleation has to occur immediately after contact before the salt particle dissolves in the solution droplet. There is no immersion mode setup conceivable to compare the freezing efficiencies. Therefore it cannot be concluded whether the collision itself is responsible for freezing. In the experiment by Yang et al., no collision was involved in ice nucleation but a movement of the three-phase contact line when the droplet adjusted to the changing electric field.*

pg. 13, line3 28-29: "One reason for this may be that in CLINCH and IMCA/ZINC experiments the particles are free to realize the energetically most favorable position in or on the droplet." This may be true, but it is also true for many of the other experiments noted above.
*Indeed, the experimental conditions have to be considered in detail. In case of AgI, a detailed discussion is given in the companion paper.*

pg. 14, line 5: "Own observations..." I think you mean to start that sentence

with "Our".

*Thank you for pointing this out.*

Table A1: Include $N$.

*We add $N$ in the revised manuscript.*

**Bibliography**

[1] N. Hoffmann, D. Duft, A. Kiselev, and T. Leisner. Contact freezing efficiency of mineral dust aerosols studied in an electrodynamic balance: quantitative size and temperature dependence for illite particles. *Faraday Discuss.*, 165:383 – 390, 2013.

[2] J. Niehaus, K.W. Bunker, S. China, A. Kostinski, C. Mazzoleni, and W. Cantrell. A technique to measure ice nuclei in the contact mode. *J. Atmos. Ocean. Technol.*, 71:3659 – 3667, 2014.

[3] B. Nagare, C. Marcolli, O. Stetzer, and U. Lohmann. Comparison of measured and calculated collision efficiencies at low temperatures. *Atmos. Chem. Phys.*, 15(23):13759 – 13776, 2015.

[4] N. Gokhale and J. Goold, Jr. Droplet freezing by surface nucleation. *J. Appl. Meteorol.*, 7:870 – 874, 1968.

[5] R. Sax and P. Goldsmith. Nucleation of water drops by brownian contact with AgI and other aerosols. *Q. J. Roy. Meteorol. Soc.*, 98:60 – 72, 1972.

[6] N. Gokhale and J. Spengler. Freezing of freely suspended, supercooled water drops by contact nucleation. *J. Appl. Meteorol.*, 11:157 – 160, 1972.

[7] N. Gokhale and O. Lewinter. Microcinematographic studies of contact nucleation. *J. Appl. Meteorol.*, 10:469 – 473, 1971.

[8] R. Pitter and H. Pruppacher. A wind tunnel investigation of freezing of small water drops falling at terminal velocity in air. *Q. J. Roy. Meteorol. Soc.*, 99:540 – 550, 1973.

[9] J. Niehaus, J. G. Becker, A. Kostinski, and W. Cantrell. Laboratory measurements of contact freezing by dust and bacteria at temperatures of mixed-phase clouds. *J. Atmos. Sci.*, 71(10):3659 – 3667, 2014.

[10] A. J. Durant and R. A. Shaw. Evaporation freezing by contact nucleation inside-out. *Geophys. Res. Lett.*, 32(20):L20814, 2005.

[11] R. D. Davis, S. Lance, J. A. Gordon, S. B. Ushijima, and M. A. Tolbert. Contact efflorescence as a pathway for crystallization of atmospherically relevant particles. *PNAS*, 112(52):15815 – 15820, 2015.

[12] J. Niehaus and W. Cantrell. Contact freezing of water by salts. *J. Phys. Chem. Lett.*, 6(17):3490 – 3495, 2015.

[13] F. Yang, R. A. Shaw, C. W. Gurganus, S. K. Chong, and Y. K. Yap. Ice nucleation at the contact line triggered by transient electrowetting fields. *Applied Physics Letters*, 107(26):264101, 2015.

---

## Author Comment (AC2) · 24 May 2016

*We thank Alexei Kiselev and Nadine Hoffmann for their careful reading of the manuscript and the suggestions for improvement. We address the points raised by them below (in italic).*

The manuscript of Nagare et al. describes contact freezing experiments conducted with supercooled droplets freely falling through the chamber containing ice nucleating (IN) aerosol particles. At the first glance, this seems to be the ideal experimental setup to study the evasive phenomena of contact freezing. The temperature regime, humidity, droplet and particle size are all atmospheric relevant.

Already the second glance reveals the difficulties in measuring the magnitudes in question (collision efficiency, CE, and freezing efficiency, FE) and interpreting the observation results. Most puzzling, the authors report no enhancement of contact freezing compared to the immersion freezing conducted with the same IN particles and in similar setup. We share the attitude of the Reviewer 1 that the manuscript should be published to provide the basis for the discussion of the presented measurements and data interpretation. There are some issues, however, that could be improved at the stage of preparation of the final manuscript that we would like to discuss here.

To our opinion, the main conclusion about the role of contact freezing made in this paper is a direct consequence of the method used to measure the CE and the approach used to compare the contact and immersion freezing behavior. We discuss these issues below followed by more specific remarks.

1. The uncertainty in interpretation of contact freezing results is obviously related to the uncertainty of determination of collision efficiency. The CE experimentally determined for 0.2 µm AgI particles (0.13) was reported being 14 times larger than calculated theoretical value. What could be the reason for such a high discrepancy? The degree of control of the experimental parameters (temperature, humidity, size and evaporation rate of the droplets, particle number concentration, droplet charge etc.) is very high, and all known interaction forces seem to be taken into account. On the other hand, the sensitivity of experimental observables to the value of the CE is very strong (see equation 19 and discussion of figure 4 in (Nagare et al., 2015), so that its knowledge is crucial for drawing a conclusion about the role of the contact freezing.

*We discussed collision efficiency in Nagare et al. (2015) and concluded that the discrepancy should come from phoretic forces that seem to be not well constrained at low temperature. Indeed, we presented in Nagare et al. (2015) the first dataset of CE acquired at sub-zero temperatures.*

One of the possible explanations would be the depletion of the IN particle concentration within the volume swept by the droplet train. Water droplets generated with 100 Hz frequency and falling with 0.186 m/s terminal velocity would be separated by 2 mm distance or 0.01 s time lag. The RMS diffusion displacement calculated for 0.2 µm diameter particle (Hinds, 1999, equation 7.18) is about 2e-6 m. Assuming the CE for 0.2 µm AgI particles equal to 0.13, the radius of the cylindrical volume where a falling 80 µm droplet experiences collisions with aerosol particles is

$$r_{drop} \times (CE)^{0.5} = 4 \times 10^{-5} m \times (0.13)^{0.5} = 1.4 \times 10^{-5} m,$$

seven times larger than the RMS Brownian displacement of the AgI particle within the droplet inter-arrival time. This essentially means that the particle number concentration reduced due to the scavenging by falling droplet will not return to equilibrium before the next droplet arrives. Reduced number concentration has to be compensated by higher apparent freezing efficiency (according to equation 19 from (Nagare et al., 2015) to describe the observed fraction of frozen droplets. This back-of-the-envelope calculation shows that the depletion of aerosol in the droplet train zone is quite possible and might affect the calculation of the collision efficiency. For larger aerosol particles the depletion can be even larger.

*We discussed CE in depth in Nagare et al. (2015). We concluded that the distance between droplets should be large enough to avoid any interference. If depletion of IN particles between droplets occurred, the injected concentration of aerosol particles would be higher than the one experienced by the droplet. This means that we insert a too high concentration into Eq. 19 from Nagare et al. (2015). When we insert the correct lower concentration into Eq. 19, CE would need to be even higher to realize the measured FF. This would not decrease but increase the difference between measured and calculated CE.*

2. We share the confusion of reviewer 1 with respect to the discussion of freezing efficiency calculated with equation 1 or equation 4 (Section 4.1). In your preceding paper (Nagare et al., 2015) the CE has been calculated with equation 19, which is just equation 4 of this manuscript under assumption that FE = 1. To my understanding, with the CE defined in this way the FE should be derived using equation 4 and not with the equation 1. It is correct that the time independence of FE should indicate the contact freezing but it has nothing to do with the number of collisions required to induce the freezing, it can happen on the first collision or after several dozens of them and still have to be the dominant mechanism. Surely the choice of equation could not be helpful to decide which freezing mechanism is dominating the apparent freezing rate of the droplets.

*Eq. (4) applies when the first of several collisions leads to freezing. This is the case for the plateau condition that we used in Nagare et al. (2015) to derive CE for AgI. However, this is not always the case. With other setups one can observe the number of collisions that is needed for freezing (e.g. Hoffmann et al., 2013a; 2013b; Niehaus et al. 2014). If indeed several collisions are needed to induce freezing, Eq (1) leads to a more accurate number for CE. Therefore, one has to decide from case to case whether Eq. (1) or (4) applies. This introduces an additional uncertainty in the derivation of FE. See also answer to reviewer 1. Since reviewers 1 and 2 were confused by our procedure to derive FE, we improve the text in the revised manuscript by adding before Eq. (1): "If a droplet freezes after more than one particle hit it, it is not clear which particle induced freezing. Assuming that all collisions were needed for freezing leads to the following equation:"*

3. Why is the freezing induced by a particle adhering to the surface of the droplet called "freezing inside-out" throughout the paper? In the original paper (Durant and Shaw, 2005) the IN particle was penetrating the surface from inside of an evaporating droplet, hence the name. In the present manuscript this name is used to describe the situation where an IN particle adheres to the surface and is only partly immersed into the droplet, as compared to the fully immersion freezing mode in IMCA/ZINC. To our understanding, partial immersion does not imply a new nucleation mechanism different. As have been shown in (Hoffmann et al., 2013a), the contact freezing efficiency of mineral dust IN particles is proportional to the surface area of the particle. We argue there that the term "contact freezing" does not imply freezing on a point contact but a considerable fraction of particle surface has to be involved into the freezing process. The term "freezing inside-out" was used to highlight the process of penetration of the droplet surface, and is not fully applicable in the present manuscript. Please consider removing this term from the paper.

*By calling freezing induced by a particle adhering to the surface "contact freezing inside-out", we wanted to discriminate it from collisional contact freezing and emphasize the similarity to contact freezing inside-out. While in the experiments performed by Durant and Shaw, 2005 the position of the particle with respect to the droplet was fixed, in our experiment the particle is free to take the energetically most favorable position in or on the droplet. Similar to Durant and Shaw, we claim that the position on the surface may be able to induce freezing at a higher temperature than if the particle is totally immersed. We recognize the difference to "contact freezing inside-out" and will therefore call it "adhesion freezing" in the revised manuscript.*

4.  Based on the comparison of FF and freezing onset temperatures you conclude that the contact freezing is a not dominant freezing process. This conclusion seems questionable, because to our opinion the true value of a contact FE for AgI cannot be derived from the experiment. A better way to compare the two freezing process is based on their characteristic times, as suggested in (Hoffmann et al., 2013b). There we have introduced a characteristic residence time $t_{im}$ of a supercooled droplet experiencing collisions with the IN particles as

$$t_{im} = \frac{2 \cdot FE_{contact}}{J_{imm}},$$

where $J_{imm}$ is the rate of freezing due to immersion freezing and can be estimated from the IMCA/ZINC measurements using the relationship between the number of unfrozen and total number of droplets:

$$\frac{N_{unfrozen}}{N_{total}} = 1 - FF = \exp(-J_{imm} \cdot t)$$

and the residence time in ZINC of $t = 3\,s$. At T = 255K the FF for the AgI 0.2 µm particles in the immersion freezing experiment (we refer to figure 1) is ≈0.95 for number concentration of 5000 cm$^{-3}$ and therefore $J_{imm} = 1s^{-1}$. The meaning of this is that on average, droplets would freeze in immersion mode 1 second after collision with 200 µm AgI particle at this temperature. *We had expected the same but inspection of Fig. 4 shows that this is not the case: For both residence times and all concentrations but the highest one, FE is below the detection limit in CLINCH while it is around 0.95 in IMCA/ZINC. This led us to have a closer look at heterogeneous ice nucleation with AgI. See companion paper: Marcolli C., Nagare, B., Welti, A., and Lohmann U.: Ice nucleation efficiency of AgI: review and new insights, Atmos. Chem. Phys. Discuss., doi:10.5194/acp-2016-142, 2016.*

Even if we assume the $FE_{contact} = 1$ (in the figure 4 it is rather 0.3 to 0.5) the characteristic time would be $t_{im} \approx 2s$, comparable to the shortest residence time used in your experiment, and thus the condition

$$FE_{contact} \gg \frac{1}{2} J_{imm} t$$

required to observe the dominance of contact freezing is not fulfilled. This simple analysis show that at least for silver iodide particles both freezing mechanisms are competing and there is no way to derive the FE for both mechanisms separately on the time scale of the experiment.

As it is immediately follows from these considerations, the freezing on-set in purely immersion mode should be always observed at higher temperature just due to the fact, that in INCA/ZINC droplets are entering the cold zone carrying the IN particles inside, whereas time is needed in CLINCH for droplet first to collect an IN particle and then freeze due to one of the freezing mechanisms.

*We think that because of the uncertainties to derive FE from FF in CLINCH, we need a clear difference in onset temperatures instead of just a difference in nucleation rate to state that either immersion or contact freezing is more efficient. Also, in contrast to the reviewers reasoning, the onset temperature of freezing for kaolinite in IMCA/ZINC was lower than in CLINCH.*

*We do not derive FE for specific mechanisms but for the two different instruments CLINCH and IMCA/ZINC. So we have an FE for the CLINCH and an FE for the IMCA/ZINC experiments. We needed further analysis, e.g. the comparison of the FE for 2s and 4s residence times in CLINCH to reach conclusions concerning the freezing mechanisms. If FE observed for the IMCA/ZINC experiment with 3 s residence time and FE for the CLINCH experiment with 4 s residence time are similar, this indicates that freezing in CLINCH might occur in immersion mode after the droplet has captured the particle. To discriminate between collisional contact freezing and immersion freezing we compared FE(2s) and FE(4s).*

5. Difference or equality of FE at different reference times is discussed throughout the manuscript. However, the difference in FE for different aerosol concentrations (as seen in the figure 4) is neglected. This behavior cannot be explained by interplay of the immersion vs. contact freezing as it is done for the residence time dependence.

*Only for AgI aerosols, experiments with different particle concentrations could be performed due to experimental reasons. Our analysis in Nagare et al. (2015) showed that at temperatures for which the plateau condition is valid, freezing efficiencies for all concentration are the same within experimental error. At temperatures for which the plateau condition is not valid, freezing efficiencies were higher for experiments where the number of collisions N > 1. This is discussed in the paper.*

**Specific comments**

1. There is an apparent contradiction between two statements (page 6, lines 21-25).: "*Panel (a) of Fig. 4 shows that FE does not exceed 0.5 for C =5000 cm−3 because it is assumed that on average 2.35 collisions are necessary to freeze a droplet. This led us to Eq. (4) to calculate FE, which assumes that already the first collision induces droplet freezing*". If more than one collision is needed to freeze the droplet, the freezing could occur on any of the subsequent collisions, couldn't it?

*Yes, this introduces indeed an additional uncertainty when the number of collisions in the chamber exceeds 1. To make this clearer in the revised manuscript, we now write on pages 6/7: "For this concentration and residence time 2.35 collisions occurred in the chamber and freezing might have been induced by any of these collisions. Eq. (1) assumes that indeed all collisions are necessary to freeze a droplet and gives a lower limit of freezing efficiency. An upper limit is obtained using Eq. (4), which assumes that the first collision induces freezing."*

2. (page 6, lines 25-26).: "*This reinforces the assumption that the first contact leads to droplet freezing in this temperature range and confirms the plateau condition used in Nagare et al. (2015) to derive CE.*" This is a confusing statement: in your previous paper the CE was derived from the measurements of FF under assumption that FE = 1 (in the plateau region, at T < 245K), and now you derive the FE value from essentially the same measurements under assumption of known CE? In this case you can't obtain any other value of FE but 1.

*Yes, indeed. This result is expected. It just shows the consistency of the approach.*

3. (page 10, line 28) Could you clarify why is the solubility of silver iodide important for the contact freezing experiments reported in this paper and what do mean by the statement "*Moreover the*

*freezing  ability depends on the surface charge on AgI particles*" (page 11 line 1).

*For an explanation of this, we refer to the companion paper (Marcolli et al., 2016), which has been published in ACPD. This paper is a review of the ice nucleation ability of AgI and discusses the questions raised by the reviewer in detail.*

**References**

Durant, A. J. and Shaw, R. A.: Evaporation freezing by contact nucleation inside-out, Geophys. Res. Lett., 32(20), 2–5, doi:10.1029/2005GL024175, 2005.

Hinds, W. C.: Aerosol technology: properties, behavior, and measurement of airborne particles, Wiley Interscience, 1999.

Hoffmann, N., Duft, D., Kiselev, A. and Leisner, T.: Contact freezing efficiency of mineral dust aerosols studied in an electrodynamic balance: quantitative size and temperature dependence for illite particles, Faraday Discuss., 165(0), 383, doi:10.1039/c3fd00033h, 2013a.

Hoffmann, N., Kiselev, A., Rzesanke, D., Duft, D. and Leisner, T.: Experimental quantification of contact freezing in an electrodynamic balance, Atmos. Meas. Tech., 6(9), 2373–2382, doi:10.5194/amt-6-2373- 2013, 2013b.

Nagare, B., Marcolli, C., Stetzer, O. and Lohmann, U.: Comparison of measured and calculated collision efficiencies at low temperatures, Atmos. Chem. Phys., 15(23), 13759–13776, doi:10.5194/acp-15-13759- 2015, 2015.

---

## Author Comment (AC3) · 24 May 2016

**Response to Anonymous Referee #3**

*We would like to thank the reviewer for careful reading the manuscript and the suggestions of improvement of readability. The responses to the comments and questions are given below in italic.*

My major comment is that main conclusions of the paper are not visible. Some revision is needed to enhance the readability, and also it is necessary to clarify what are the major conclusions of the paper. It is also not clear why this study is important, and what the atmospheric implications are. Some more discussion is needed to understand why AgI, ATD and Kaolinite particles were used, why natural dust or soil dust particles were not chosen as these are more atmospherically relevant. This is nice study, overall contact freezing is not well understood, but main message is buried. Below some comments may help to revise this paper further.

*It is important to know whether contact freezing is more efficient than immersion freezing for parameterization in atmospheric models and for the microphysical understanding of the different heterogeneous ice nucleation processes. We emphasize this now more in the abstract by adding on line 4:*

*"To date, direct comparisons of contact and immersion freezing with the same INP, for similar residence times and concentrations are lacking."*

*We preferred to first study INPs that have been investigated before. This is the case for AgI, ATD, and kaolinite. Moreover, these samples represent different types of ice nuclei. ATD is a mixture of different minerals, while AgI and kaolinite contain one main component. AgI induces ice nucleation at a rather high temperature and shows a close lattice match with ice, while kaolinite does not have a close lattice match and induces freezing at much lower temperature. In future this study could be extended to investigate natural mineral dust or soil dust samples.*

-What is the typical size of supercooled droplets observed in mixed phase clouds? How often 80 um droplets are observed. Atmospheric relevance of droplet size should be discussed.

*For heterogeneous nucleation, droplet size is not important. The relevant quantity is the surface of the INP, which is in our experiments in the atmospherically relevant range.*

-Following two sentences (i and ii) needs to be elaborated. Bulk liquid water properties are different from individual water droplet properties. Please define what you mean by sprinkling. Do particles were size-selected, how many particles were used, what is the temperature of the liquid water, do water is pure or distilled or regular lab supply grade, how long this experiment was performed, do all particles sediment, and how this observation was made (visual observation, microscope).

(i) "When we sprinkled ATD on a water surface, most particles immediately immersed and sank to the bottom. This suggests that when ATD particles collide with water droplets, the particles become immediately immersed such that in immersion freezing and contact freezing experiments the immersion mode is probed."

(ii) "When we sprinkled kaolinite powder on water, we observed that some particles floated on the surface while others became totally immersed and sank to the bottom."

*We give the requested information in the revised manuscript in the new Section 2.3. These were very simple experiments to confirm the wetting behavior predicted by evaluating the contact angles between water and the particles. There should be no difference between a droplet surface and a bulk water surface as long as the Kelvin effect is not important, which is the case for droplets larger 1 µm, i.e. all cloud droplets. We add to the observations for kaolinite the timescale:"… some floated on the surface for hours while others became totally immersed and sank to the bottom within seconds."*

-It is mentioned that "A particle on the surface can induce ice nucleation in the immersion mode with the part immersed in water or in contact mode with the part exposed to air." How this can be assumed, what is the basis for this?

*We refer here to the contact freezing process occurring when a particle adheres to the surface of a droplet. In the ACPD version of the manuscript we referred to this process as contact freezing inside-out. In the revised manuscript we change the terminology to "adhesion freezing" because naming it "contact freezing inside-out" was criticized by reviewers 2. If the part of the particle that is exposed to the surface is less efficient at*

*nucleating ice than the part of the particle immersed in the droplet, the freezing efficiency should still equal approximately the one observed for cases when the particle is totally immersed in water.*

-Section 5.6: It is not clear what results are discussed. This section looks like reading a literature review. There is only one sentence (The immersion and contact freezing studies compiled in Fig. 6 suggest that contact freezing is more efficient than immersion freezing with an onset temperature that is about 3 K higher), which describes the results, but there is no discussion. I suggest use present results to discuss the figure 6, but not previous results (as they have different instrument platform to study Kaolinite properties). For example XRD analysis of Kaolinite particles differ from group to group because of the XRD instrument sensitivity issues, and also impurities within the Kaolinite samples. Note that Kaolinite from different vendors have different properties, also shown by Wex et al (http://www.atmos-chem-phys.net/14/5529/2014/acp-14-5529-2014.pdf) who shown ice nucleating properties are sensitive to the particles procured from different vendors.

*We are aware of the different qualities of kaolinite depending on the vendor. For our experiment we used Fluka kaolinite from Sigma Aldrich (K-SA). We therefore compare to studies, which also used Fluka kaolinite. This is the case for Wex et al. (2014) and Tobo et al. (2012). This comparison is therefore justified. It is well known that Fluka kaolinite is not pure. This is why the composition determined by XRD is important. The information given in this section is relevant for the interpretation of the results. Reviewer 1 even asked us to "place our results more clearly in the context of previous work".*

-Section 5.6: Second paragraph. How this is applicable to the present study. This material is not relevant, if yes please discuss how. As mentioned above this reads like a literature review.

*We need information about the morphology and surfaces of kaolinite to discuss whether kaolinite particles adhere to the surface of the droplet or whether they are immersed. A discussion of previous literature is needed. Reviewer 1 even asked for a more profound discussion of kaolinite and suggested inclusion of more previous work.*

-Please see Section 5.5 too. Discuss the present results. There is lot of discussion on previous studies, but how they are related to this study. It is not clear why these studies are discussed. I suggest move this material to Intro section to increase the readability.

*In this section the results for ATD are discussed and put in context with previous studies on ATD. Such a discussion is necessary.*

-Last three sentences from Conclusion section (page 14, line 8-11). Do authors performed any experiments to conclude this, or these are the conclusions from previous studies. If later then I suggest move this to intro section.

*These are the conclusions of the present study. We make this clearer in the revised manuscript by writing: "Our experiments and calculations…"*

-Can majority of Section 5.2 (except page 17, line 17-23) and Section 5.3 be moved to Intro section? They do not discuss any results.

*We moved Section 5.2 to the introduction and Section 5.3 to an appendix.*

-It may be a good idea to combine section 4 and 5. Section 5, for dust particles, has lot of discussion concerning previous studies and may help to increase the readability.

*We prefer to keep the results and the discussion of the results apart.*